# Technical note: Calculation scripts for ensemble hydrograph separation

James W. Kirchner[1,2,3] and Julia L.A. Knapp[1]

[1]Dept. of Environmental Systems Science, ETH Zurich, Zurich, Switzerland
[2]Swiss Federal Research Institute WSL, Birmensdorf, Switzerland
[3]Department of Earth and Planetary Science, University of California, Berkeley, California, USA

*Correspondence to*: James Kirchner (kirchner@ethz.ch)

**Abstract.** Ensemble hydrograph separation has recently been proposed as a technique for using passive tracers to measure catchment transit time distributions and new water fractions, introducing a powerful new tool for quantifying catchment behavior. However, the technical details of the necessary calculations may not be straightforward for many users to implement. We have therefore developed scripts that perform these calculations on two widely used platforms (MATLAB and R), to make these methods more accessible to the community. These scripts implement robust estimation techniques by default, making their results highly resistant to outliers. Here we briefly describe how these scripts work, and offer advice on their use. We illustrate their potential and limitations using synthetic benchmark data.

## 1. Introduction

What fraction of streamflow is composed of recent precipitation? Conversely, what fraction of precipitation becomes streamflow promptly? What is the age distribution of streamwater? What is the "life expectancy" of precipitation as it enters a catchment? And how do all of these quantities vary with catchment wetness, precipitation intensity, and landscape characteristics? Questions like these are fundamental to understanding the hydrological functioning of landscapes and characterizing catchment behavior. Ensemble hydrograph separation (EHS) has recently been proposed as a new tool for quantifying catchment transit times, using time series of passive tracers like stable water isotopes or chloride. Benchmark tests using synthetic data have shown that this method should yield quantitatively accurate answers to the questions posed above (Kirchner, 2019), and initial applications to real-world data sets (e.g., Knapp et al., 2019) have demonstrated the potential of this technique.

However, it has become clear over the past year that the equations of Kirchner (2019, hereafter denoted K2019) may be difficult for many users to implement in practically workable calculation procedures or computer codes. It has also become clear that robust estimation methods would be a valuable addition to the ensemble hydrograph separation toolkit, given the likelihood of outliers in typical environmental data sets. The present contribution is intended to fill both of these needs, by

presenting user-friendly scripts that perform EHS calculations in either MATLAB or R, and that implement robust estimation by default.

Here we demonstrate these scripts using synthetic data generated by the benchmark model of K2019, which in turn was adapted from the benchmark model of Kirchner (2016). We use these benchmark data instead of real-world observations, because age-tracking in the model tells us what the correct answers are, so that we can verify how accurately these EHS scripts infer water ages from the synthetic tracer time series. The benchmark model consists of two non-linear boxes coupled in series, with a fraction of the discharge from the upper box being routed directly to streamflow, and the rest being routed to the lower box, which in turn discharges to streamflow (for further details, see Kirchner, 2016 and K2019). It should be emphasized that the benchmark model and the ensemble hydrograph separation scripts are completely independent of one another. The benchmark model is not based on the assumptions that underlie the ensemble hydrograph separation method. Likewise, the EHS scripts do not know anything about the internal workings of the benchmark model; they only know the input and output water fluxes and their isotope signatures. Thus the analyses presented here are realistic analogues to the real-world problem of trying to infer the internal functioning of catchments from only their inputs and outputs of water and tracers.

Figures 1a and 1b show the simulated daily water fluxes and isotope ratios used in most of the analyses below. The precipitation fluxes are averages over the previous day (to mimic the effects of daily time-integrated precipitation sampling), and the streamflow values are instantaneous values at the end of each day (to mimic the effects of daily grab sampling). We also aggregated these daily values to simulate weekly sampling, using weekly volume-weighted average tracer concentrations in precipitation and weekly spot values in streamflow (representing grab samples taken at the end of each week). Five percent of the simulated tracer time series were randomly deleted to mimic sampling and measurement failures, and a small amount of random noise was added to mimic measurement errors.

To illustrate the need for robust estimation techniques, and to demonstrate the effectiveness of the robust estimation methods employed in our scripts, we also randomly corrupted the synthetic isotope data with outliers (Fig. 1c). These outliers are intentionally large; for comparison, the entire range of the outlier-free data shown in Fig. 1b lies between the two dashed lines in Fig. 1c. The outliers are also strongly biased (they all deviate downward from the true values), making them harder to detect and eliminate. We make no claim that the size of these outliers, and their frequency in the data set, reflect outlier prevalence and magnitude in the real world (which would be difficult to estimate in practice, without replicate sampling or other independent reference data). Instead, these outliers were simply chosen to be large enough, and frequent enough, that they will substantially distort the results of non-robust analyses. They thus provide a useful test for the robust estimation methods described below.

## 2. Estimating new water fractions using the function EHS_Fnew

The simplest form of ensemble hydrograph separation seeks to measure the fraction of streamflow that is composed of recent precipitation. Conventional hydrograph separation uses end-member mixing to estimate the time-varying contributions of "event water" and "pre-event water" to streamflow. By contrast, ensemble hydrograph separation seeks to estimate the *average* fraction of new water in streamflow, averaged over an *ensemble* of events (hence the name), based on the regression slope between tracer fluctuations in precipitation and discharge (see Fig. 2a),

$$C_{\mathrm{Q}_j} - C_{\mathrm{Q}_{j-1}} = {}^{\mathrm{Qp}}F_{\mathrm{new}} \left(C_{\mathrm{P}_j} - C_{\mathrm{Q}_{j-1}}\right) + \alpha + \varepsilon_j \quad , \tag{1}$$

where ${}^{\mathrm{Qp}}F_{\mathrm{new}}$ is the "event new water fraction" (the average fraction of new water in streamflow during sampling intervals with precipitation), $C_{\mathrm{Q}_j}$ and $C_{\mathrm{Q}_{j-1}}$ are the tracer concentrations in streamflow at time steps $j$ and $j-1$, $C_{\mathrm{P}_j}$ is the volume-weighted average tracer concentration in precipitation that falls between time $j-1$ and time $j$, and the intercept $\alpha$ and the error term $\varepsilon_j$ can be viewed as subsuming any bias or random error introduced by measurement noise, evapoconcentration effects, and so forth (see Sect. 2 of K2019 for formulae and derivations).

Although ensemble hydrograph separation is rooted in assumptions that are similar to end-member mixing, mathematically speaking it is based on correlations between tracer fluctuations rather than on tracer mass balances. As a result, it does not require that the end-member signatures are constant over time, or that all the end-members are sampled or even known, and it is relatively unaffected by evaporative isotopic fractionation or other biases in the underlying data (see Sect. 3.6 of K2019). Even when new water fractions are highly variable over time, one can show mathematically (and confirm with benchmark tests) that ensemble hydrograph separation will accurately estimate their average (see Sect. 2 and Appendix A of K2019). As Fig. 2a shows, higher discharges (indicating wetter catchment conditions) may be associated with larger new water fractions, and thus stronger coupling between tracer fluctuations in precipitation and streamflow. Nonetheless, the regression slope in Fig. 2a averages over these variations, yielding an event new water fraction (0.164±0.006) that equals, within error, the true event new water fraction (0.168±0.005) determined by age tracking in the benchmark model.

The lagged streamflow tracer concentration $C_{\mathrm{Q}_{j-1}}$ serves as a reference level for measuring the fluctuations in the tracer concentrations $C_{\mathrm{P}_j}$ and $C_{\mathrm{Q}_j}$ in time step $j$. This has the practical consequence that the sampling interval determines what "new water" means. For example, if $C_{\mathrm{P}}$ and $C_{\mathrm{Q}}$ are sampled daily, "new water" means water that fell within the previous day (and thus is expressed in units of day$^{-1}$), and if they are sampled weekly, "new water" means water that fell within the previous week (and thus is expressed in units of week$^{-1}$). Because the meaning and dimensions of new water fractions depend on the sampling interval, so do the numerical values, as illustrated by Knapp et al. (2019). In our example, the weekly event new water fraction, calculated from weekly sampling of the daily values in Fig. 1, is 0.443±0.024, which agrees, within error, with the true weekly event new water fraction (0.429±0.017). Astute readers will notice that the weekly

new water fraction is not 7 times the daily one, implying that translating between weekly and daily event new water fractions is not just a matter of converting the units. This is partly because weeks rarely consist of seven consecutive daily hydrological events (instead they typically include some days without rain). Thus the relationship between daily and weekly new water fractions will depend on the intermittency of precipitation events. One must also keep in mind that the proportion of new water in streamflow cannot exceed 1, so new water fractions, even when evaluated from low-frequency data, cannot be arbitrarily large.

As explained in Sect. 2 of K2019, there are three main types of new water fractions. First, as noted above, the event new water fraction ${}^{Qp}F_{\mathrm{new}}$ is the average fraction of new water in streamflow during sampling intervals with precipitation. Second, the new water fraction of discharge ${}^{Q}F_{\mathrm{new}}$ is the average fraction of new water in streamflow during all sampling intervals, with or without precipitation; this will obviously be less than the event new water fraction because periods without precipitation will not contribute any new water to streamflow. Third, the "forward" new water fraction, or new water fraction of precipitation ${}^{P}F_{\mathrm{new}}$, is the average fraction of precipitation that will be discharged to streamflow within the current sampling interval. Both ${}^{Q}F_{\mathrm{new}}$ and ${}^{P}F_{\mathrm{new}}$ can be derived by re-scaling ${}^{Qp}F_{\mathrm{new}}$ from Eq. (1) by the appropriate denominators. All three of these new water fractions can also be volume-weighted (to express, for example, the fraction of new water in an average liter of streamflow, rather than on an average day of streamflow), if the regression in Eq. (1) is volume-weighted; these volume-weighted fractions are denoted using an asterisk, as ${}^{Qp}F^{*}_{\mathrm{new}}$, ${}^{Q}F^{*}_{\mathrm{new}}$, and ${}^{P}F^{*}_{\mathrm{new}}$.

In our scripts, new water fractions are calculated by the function EHS_Fnew. Users supply EHS_Fnew with vectors of evenly spaced data for the water fluxes $P$ and $Q$, and tracer concentrations $C_{\mathrm{P}}$ and $C_{\mathrm{Q}}$, in precipitation and discharge. Users can also specify five options: a) the threshold precipitation rate $p_threshold$ (in the same units as $P$), below which precipitation inputs will be ignored, under the assumption that they will mostly have been lost to canopy interception, b) $vol_wtd$, a logical flag (default=false) indicating whether the new water fractions should be volume-weighted, c) $robust$, a logical flag (default=true) indicating whether the new water fractions should be calculated using robust estimation methods as described in Sect. 2.1 below, d) $ser_corr$, a logical flag (default=true) indicating whether the standard error estimates should account for serial correlation in the residuals, and e) $ptfilter$, a point filter vector of logical flags indicating whether individual time steps should be included in the analysis, thus facilitating easy analyses of subsets of the original time series. The function EHS_Fnew returns estimates of ${}^{Qp}F_{\mathrm{new}}$, ${}^{Q}F_{\mathrm{new}}$, and ${}^{P}F_{\mathrm{new}}$ and their associated standard errors, with or without volume-weighting depending on whether $vol_wtd$ is set to true or false.

### 2.1 Robust estimation of new water fractions

The linear regression in Eq. (1), like any least-squares technique, is potentially vulnerable to outliers. Because potential outliers are often present in environmental data, practical applications of ensemble hydrograph separation would benefit from a robust method for estimating new water fractions. Such a method should not only be insensitive to outliers; ideally it should also be statistically efficient (i.e., it should yield reasonable estimates from small samples), and it should be asymptotically unbiased (i.e., it should converge to the conventional regression results when outliers are absent, with a bias near zero for large samples).

Figure 2 shows ensemble hydrograph separation plots of the outlier-free benchmark data (Fig. 2a, estimated from the time series shown in Fig. 1b) and the outlier-corrupted benchmark data (Fig. 2b, estimated from the time series shown in Fig. 1c). On these axes – precipitation and streamflow tracer fluctuations on the $x$ and $y$ axes, respectively, each expressed relative to the streamflow tracer concentration in the previous time step – the regression slope estimates the event new water fraction ${}^{\mathrm{Qp}}F_{\mathrm{new}}$. Here we are interested in how outliers affect this regression slope. When outliers are absent (Fig. 2a), the regression slope (0.164±0.006, estimate±standard error) is consistent with the true event new water fraction ${}^{\mathrm{Qp}}F_{\mathrm{new}}$= (0.168±0.005) calculated from water age tracking in the benchmark model.

By contrast, outliers substantially distort the ensemble hydrograph separation plot in Fig. 2b; they extend well beyond the range of the outlier-free data indicated by the gray rectangle, and inflate the estimate of ${}^{\mathrm{Qp}}F_{\mathrm{new}}$ by nearly a factor of three. Outliers in precipitation tracer concentrations will be displaced left or right from the corresponding true values (in Fig. 2b, these outliers are displaced to the left because they are all negative). Precipitation outliers will thus tend to flatten the regression line. Outliers in streamflow concentrations will appear in two different ways. First, they will be displaced above or below the corresponding true values (in this case, they are only displaced below, because they are all negative). Secondly, they will also appear as strongly correlated deviations on both the $x$ and $y$ axes because streamflow concentrations at time $j$-1 are used as reference values for both precipitation concentrations (on the $x$ axis) and streamflow concentrations (on the $y$ axis) at time $j$. Unlike precipitation outliers, these correlated points will tend to artificially steepen the regression line. Thus, whether outliers steepen or flatten the regression relationships underlying ensemble hydrograph separation will depend on the relative abundance and size of the streamflow outliers and precipitation outliers (relative to each other, and relative to the variability in the true streamflow and precipitation tracer values). In the example shown in Fig. 2b, the outliers have the net effect of artificially steepening the fitted slope, yielding an apparent ${}^{\mathrm{Qp}}F_{\mathrm{new}}$ of 0.430±0.018 that is more than 2.5 times the true value of 0.168±0.005 determined by age tracking in the benchmark model.

Many robust estimation methods will not be effective against outliers like those shown in Fig. 2b, which create points that have great leverage on the slope of the fitted line. This leverage can allow the outliers to pull the line close enough to

themselves that they will not be readily detected as outliers. To address this problem, our robust estimation procedure has two parts. The first step is to identify extreme values of both precipitation and streamflow tracer concentrations at the outset, and exclude them by setting them to NA (thus treating them as missing values). This will effectively prevent outliers from exerting strong leverage on the solution. Because the exclusion criterion must itself be insensitive to outliers, we define extreme values as those that lie farther from the median than six times MAD, the median absolute deviation from the median. The cutoff value of six times MAD was borrowed from the residual downweighting function used in Locally Weighted Scatterplot Smoothing (LOWESS: Cleveland, 1979). Any exclusion criterion may also eliminate points that are not outliers, but simply extreme values. However, unless the underlying distribution has very long tails, the $6 \cdot \mathrm{MAD}$ criterion will exclude very few points that are not outliers. If the underlying data follow a normal distribution, for example, the chosen criterion will exclude only the outer-most 0.005 percent of that distribution.

As a second step, we use iteratively reweighted least squares (IRLS: Holland and Welsch, 1977) to estimate the regression slope, and thus the event new water fraction ${}^{\mathrm{Qp}}F_{\mathrm{new}}$. IRLS iteratively fits Eq. (1) by linear regression, with point weights that are updated after each iteration. Points with unusually large residuals are given smaller weight. In this way, IRLS regressions follow the linear trend in the bulk of the data, giving less weight to points that deviate substantially from that trend. This behavior, which allows IRLS to down-weight outliers, can have undesirable effects in analyses of outlier-free data exhibiting divergent trends. In Fig. 2a, for example, higher flows have steeper trends, with the highest 20 percent of flows (shown in red) exhibiting a much steeper trend than the rest of the data. Because IRLS gives these points relatively less weight, the robust estimate of ${}^{\mathrm{Qp}}F_{\mathrm{new}}$ is 0.126±0.004, 25% less than the true value of 0.168±0.005 determined from age tracking in the benchmark model. Thus, in this case, the robust estimation procedure is somewhat less accurate than ordinary least squares if the data are free of outliers. Conversely, however, the outliers in Fig. 2b have little effect on the robust estimation procedure, which returns a ${}^{\mathrm{Qp}}F_{\mathrm{new}}$ estimate of 0.115±0.005, within 10% of the outlier-free value. This example demonstrates that like any robust estimation procedure, ours is highly resistant to outliers but at the cost of reduced accuracy when outliers are absent, particularly in cases, like Fig. 2a, that superimpose widely differing trends. Robust estimation is turned on by default, but users can turn it off if they are confident that their data are free of significant outliers.

## 3. Profiling catchment behavior using EHS_profile

Visual comparison of the different discharge ranges shown by different colors in Fig. 2a indicate that in these benchmark data, higher discharges are associated with stronger coupling between tracer concentrations in precipitation and streamflow, implying that streamflow contains a larger fraction of recent precipitation. This observation implies that by estimating ${}^{\mathrm{Qp}}F_{\mathrm{new}}$ by regression for each discharge range separately, one can profile how new water fractions vary with discharge (and

thus, at least in the benchmark model system, with catchment wetness). As outlined in K2019, this can be accomplished by splitting the original data set into separate ensembles and running EHS_Fnew on each ensemble individually.

Although this can be achieved by applying a series of point filter vectors to isolate each ensemble, here we provide a function, EHS_profile, that automates this process. Users supply EHS_profile with the same data vectors and logical flags needed for EHS_Fnew as described in Sect. 2 above, plus a criterion vector for sub-setting the data and two vectors that define the percentile ranges of this criterion variable to be included in each subset. Many different variables could be chosen as the sub-setting criterion; examples include discharge (or antecedent discharge), precipitation intensity (or antecedent precipitation), day of year, soil moisture, groundwater levels, fractional snow cover, and so forth.

Figures 3 and 4 show example profiles created by EHS_profile from the benchmark model time series, with and without outliers. The gray lines in Fig. 3 show how new water fractions (the fractions of streamflow that entered the catchment as precipitation during the same sampling interval, as determined by age tracking in the benchmark model) vary as a function of discharge rates. The gray lines in Fig. 4 show the similar age tracking results for "forward" new water fractions (the fractions of precipitation that leave as streamflow during the same sampling interval), as a function of precipitation rates. These age tracking results are compared to profiles of the new water fraction ${}^{Q}F_{\mathrm{new}}$ and "forward" new water fraction ${}^{P}F_{\mathrm{new}}$ calculated from the tracer time series using EHS_profile, with and without robust estimation (dark and light symbols, respectively, in Figs. 3 and 4). If the tracer time series contain no outliers (Figs. 3a and 4a), both the robust and non-robust estimation procedures accurately estimate the new water fractions in each discharge range (i.e., the light and dark blue points closely follow the gray line). By contrast, if the tracer time series are corrupted by outliers (Figs. 3b-f and 4b), the non-robust estimation procedure yields new water fractions (light blue points) that deviate dramatically from the age tracking results, even if outliers make up only 1 percent of the data set (Fig. 3b). By contrast, the robust estimation procedure yields new water fractions (dark blue points) that closely follow the age tracking results (Figs. 3b-e and 4b), at least as long as the fraction of outliers in the data set does not exceed10 percent. Somewhere between an outlier frequency of 10 and 20 percent, the robust estimation procedure reaches its so-called "breakdown point" (Hampel, 1971), at which it can no longer resist the outliers' effects (see Fig. 3f). This breakdown point is relatively low (for comparison, the breakdown point of the median as an estimator of central tendency is 50 percent) because the outliers introduce highly correlated artifacts into the analysis (see Fig. 2b) and because these particular outliers are very large and very strongly biased (they always lie below the true values). The breakdown point could be raised by tailoring the exclusion criterion (step 1 in our two-step procedure) to these particular outlier characteristics – for example, by basing it on deviations relative to the median of the densest 50% of the data, rather than the median of all the data, to counteract the bias in the outliers. Doing so, however, would violate the principle that the scripts and the data used to test them should be fully independent of one another, as outlined in Sect. 1. In any case, the empirical breakdown point of 10-20 percent identified in Fig. 3 is specific to this particular data set with these particular outlier characteristics, and should not be interpreted as indicating the likely breakdown point in other situations. In

general, however, we would expect the robust estimation procedure to be more resistant to outliers that are smaller or less strongly biased.

Astute readers will note that the robust estimates of new water fractions almost exactly match the benchmark age tracking data in the profiles shown here, whereas they underestimated the same age tracking data by roughly 25% in Sect. 2.1 above, where the data were not separated into distinct ranges of discharge or precipitation rates. The difference between these two cases is illuminating. Individual discharge ranges exhibit well-defined relationships between tracer fluctuations in precipitation and streamflow; that is, the individual colored discharge ranges in Fig. 2a show roughly linear scatterplots with well constrained slopes. Thus for these individual discharge ranges, the robust estimates agree with the benchmark "true" values (and the non-robust estimates do too, if the underlying data are free of outliers). However, when these different discharge ranges are superimposed, the robust estimation procedure down-weights the high-discharge points because they follow a different trend from the rest of the data, resulting in an underestimate of the new water fraction averaged over all discharges. Thus users should be aware that our robust estimation procedure (like any such procedure) can be confounded by data in which some points exhibit different behavior than the rest, and are therefore excluded or down-weighted as potential anomalies.

## 4. Estimating transit time distributions using EHS_TTD

One can estimate catchment transit time distributions from tracer time series by extending Eq. (1) to a multiple regression over a series of lag intervals $k = 0 \ldots m$:

$$\left(C_{\mathrm{Q}_j} - C_{\mathrm{Q}_{j-m-1}}\right) = \sum_{k=0}^{m} \beta_k \left(C_{\mathrm{P}_{j-k}} - C_{\mathrm{Q}_{j-m-1}}\right) + \alpha + \varepsilon_j \quad , \tag{2}$$

where the vector of regression coefficients $\beta_k$ can be re-scaled to yield different types of transit time distributions as described in Sects. 4.5-4.7 of K2019. Applying Eq. (2) to catchment data is straightforward in principle but tricky in practice, because any rainless intervals will lead to missing precipitation tracer concentrations $C_{\mathrm{P}_{j-k}}$ for a range of time steps $j$ and lag intervals $k$. Handling this missing data problem requires special regression methods, as outlined in Sect. 4.2 of K2019. Gaps in the underlying data can also lead to ill-conditioning of the covariance matrix underlying the least-squares solution of Eq. (2), leading to instability in the regression coefficients $\beta_k$. This ill-conditioning problem is handled using Tikhonov-Phillips regularization, which applies a smoothness criterion to the solution in addition to the least-squares goodness-of-fit criterion, as described in Sect. 4.3 of K2019.

The function EHS_TTD spares users from the practical challenges of implementing these methods. Users supply EHS_TTD with the same data vectors needed for EHS_Fnew as described in Sect. 2 above. Users also specify $m$, the maximum lag in

the transit time distribution, and $\nu$, the fractional weight given to the Tikhonov-Phillips regularization criterion (versus the goodness-of-fit criterion) in determining the regression coefficients $\beta_k$. The default value of $\nu$ is 0.5, following Sect. 4.3 of K2019, which gives the regularization and goodness-of-fit criteria roughly equal weight; if $\nu$ is set to zero, the regularization criterion is ignored and the estimation procedure becomes equivalent to ordinary least squares. Users can set the optional point filter $Qfilter$ to filter the data set by discharge time steps (for example, to track the ages of discharge leaving the catchment during high or low flows, regardless of the conditions that prevailed when the rain fell that ultimately became those streamflows). Alternatively, users can set the optional point filter $Pfilter$ to filter the data set by precipitation time steps (for example, to track the life expectancy of rainwater that falls during large or small storms, regardless of the conditions that will prevail when that rainwater ultimately becomes discharge). It is also possible to set both $Pfilter$ and $Qfilter$ so that both the precipitation and discharge time steps are filtered, but this capability should be used cautiously because it could potentially lead to TTDs being estimated on only a small, and highly fragmented, part of the data set.

The function EHS_TTD returns vectors for the transit time distribution ${}^{Q}$TTD (the age distribution of streamflow leaving the catchment), the "forward" transit time distribution ${}^{P}$TTD (the "life expectancy" distribution of precipitation entering the catchment), and their associated standard errors. If the $vol_wtd$ flag is true, the corresponding volume-weighted distributions (${}^{Q}$TTD$^*$ and ${}^{P}$TTD$^*$) and their standard errors are returned. In all cases, the units are fractions of discharge or precipitation per sampling interval (e.g., day$^{-1}$ for daily sampling or week$^{-1}$ for weekly sampling). This difference in units should be kept in mind when comparing results obtained for different sampling intervals.

### 4.1 Robust estimation of transit time distributions

In EHS_TTD, robust estimation of transit time distributions follows a multi-step approach that is analogous to that which is used in EHS_Fnew (described in Sect. 2.1 above). We first exclude extreme values of both precipitation and streamflow tracer concentrations using the $6 \cdot \mathrm{MAD}$ criterion. We then apply iteratively reweighted least squares (IRLS) to Eq. (2), without regularization; this yields a set of robustness weights, which down-weight points that lie far away from the multidimensional linear trend of the data. These robustness weights are then applied within the Tikhonov-Phillips regularized regression that estimates the transit time distribution. This robust estimation approach requires that we handle the missing data problem in a different way than the one that was documented in Sect. 4.3 of K2019. The necessary modifications are detailed in Appendix A.

This robust estimation procedure yields transit time distributions that are highly resistant to outliers (Fig. 5). The gray lines in Fig. 5 show the true transit time distributions of discharge (${}^{Q}$TTD) and "forward" transit time distributions of precipitation (${}^{P}$TTD), as determined by age tracking in the benchmark model. These age tracking results are compared to transit time distributions calculated from the tracer time series using EHS_TTD, with and without robust estimation (dark and light

Hydrology and Earth System Sciences
EGU
Discussions

symbols, respectively, in Fig. 5). When the tracer time series are outlier-free (Fig. 5a and 5c), both the robust and non-robust estimation procedures accurately estimate these TTDs (i.e., the light and dark blue points closely follow the gray lines). When the tracer time series are corrupted by outliers (Figs. 5b and 5d), the non-robust TTDs (light blue points) deviate substantially from the age tracking results (gray lines), but the robust TTDs (dark blue points) follow the gray lines nearly as well as with the outlier-free data.

**4.2 Overestimation of uncertainties in humped transit time distributions**

The benchmark tests shown in Figs. 2-5 above, like most of those presented in K2019, are based on a benchmark model simulation that yields "L-shaped" TTDs, that is, those in which the peak occurs at the shortest lag. In this section we explore several phenomena associated with the analysis of distributions that are "humped", that is, those that peak at an intermediate lag. Where tracer data are sufficient to constrain the shape of catchment-scale TTDs, they suggest that humped distributions are rare (Godsey et al., 2010). They are also not expected on theoretical grounds, since precipitation falling close to the channel should reach it quickly and with little dispersion, leading to TTDs that peak at very short lags (Kirchner et al., 2001; Kirchner and Neal, 2013). Nonetheless, humped distributions could potentially arise in particular catchment geometries (Kirchner et al., 2001), or in circumstances where tracers are introduced far from the channel but not close to it. Thus we have re-run the benchmark model with parameters that generate humped TTDs ($S_{u,ref}$ = 50 mm, $S_{l,ref}$ = 50 mm, $b_u$ = 5, $b_l$ = 2, and $\eta$ = 0.01), driven by the same time series of precipitation rates and rainfall $\delta^{18}O$ values used in Sects. 2-4 above.

Figure 6 shows both forward and backward humped transit time distributions, as estimated by EHS_TTD from the benchmark model daily tracer time series, with their standard errors. (Here, as in the other analyses presented in this note, $ser_corr$ = true, so the standard errors account for serial correlation in the residuals.) It is visually obvious that the error bars, which represent a range of ±1 standard error, are much larger than either the differences in the TTD estimates themselves (the solid dots) between adjacent lags, or the typical deviations of the TTD estimates from the true values determined from age tracking (the gray lines). In other words, the error bars greatly exaggerate the uncertainty or unreliability of the TTD estimates. If the TTD estimates are unbiased, and their standard error estimates are too, then the standard error should approximate the root-mean-square deviation between the estimate and the benchmark. If the errors are roughly normally distributed, the true value should lie within the error bars about 65-70 percent of the time, and outside the error bars about 30-35 percent of the time. By contrast, the error bars in Fig. 6 are many times larger than the typical deviation of the TTD estimates from the true values. Figure 5 shows a milder form of this exaggeration of uncertainty; here too, the age tracking values almost never lie outside the error bars, whereas that should occur about 1/3 of the time if the error bars are estimated accurately.

Thus it appears that the TTD error estimates are generally conservative (i.e., they overestimate the true error), but with humped distributions the uncertainties are greatly exaggerated. Numerical experiments (Fig. 7) reveal that this problem

arises from the nonstationarity of the transit times in the benchmark model (and, one may presume, in real-world catchment data as well). K2019 (Sect. 4 and Appendix B) showed that ensemble hydrograph separation correctly estimates the average of the benchmark model's nonstationary TTD, as one can also see in Figs. 6 and 8. When this (stationary) average TTD is used to predict streamflow tracer concentrations (which is necessary to estimate the error variance and thus the standard errors), however, it generates nearly the correct patterns of values but not with exactly the right amplitudes or at exactly the right times (see Fig. 7a). This is the natural consequence of estimating a nonstationary process with a stationary statistical model. As a result, the residuals are larger, with much stronger serial correlations, than they would be if the underlying process were stationary (compare Figs. 7a and 7b), resulting in much larger calculated standard errors of the TTD coefficients. These tendencies are even stronger for humped TTDs, which introduce stronger serial correlations in the multiple regression fits that are used to estimate the TTD itself. Serial correlations in the residuals reduce the effective number of degrees of freedom by a factor of approximately $(1-r_{sc})/(1+r_{sc})$, where $r_{sc}$ is the lag-1 serial correlation coefficient of the residuals, thus increasing the standard error by roughly a factor of $[(1+r_{sc})/(1-r_{sc})]^{0.5}$. For the nonstationary case shown in Fig. 7a, $r_{sc}$ is 0.96, increasing the standard error by a factor of roughly 7, whereas for the stationary case shown in Fig. 7b, $r_{sc}$ is 0.22, increasing the standard error by only a factor of 1.25.

Figure 8 shows that if the same TTDs are estimated from weekly data, the standard errors more accurately approximate the mismatch between the TTD estimates and the true values (i.e., the difference between the blue dots and the gray curves). Weekly sampling yields much more reasonable standard errors in this case, because the multiple regression residuals are much less serially correlated (see Fig. 7c; $r_{sc}$ is 0.66, increasing the standard error by only a factor of 2.2). In addition, with daily data the TTD coefficients are estimated for a closely spaced mesh of lag times (with lag intervals of 1 day), and broad TTDs like the ones shown in Fig. 6 do not change much over such short lag intervals. Thus the individual TTD coefficients on such a closely spaced mesh are not well constrained; one could make the TTD stronger at the fifth daily lag and weaker at the sixth daily lag, for example, with little effect on the overall fit to the data. With weekly sampling, the TTD coefficients are more widely separated in time (with lag intervals of 1 week), and thus are less redundant with one another.

### 4.3 Overestimation of $F_{new}$ when distributions are humped

Figure 9 shows profiles of new water fractions new water fractions (${}^{Q}F_{new}$) and forward new water fractions (${}^{P}F_{new}$), analogous to those shown in Figs. 3-4, but based on model simulations yielding the humped distributions shown in Fig. 6. One can immediately see that the new water fractions are substantially overestimated, and that this bias is particularly large for forward new water fractions associated with low rainfall rates (i.e., the left side of Fig. 9b). These artifacts arise because the random fluctuations in input tracer concentrations used in the benchmark model have a serial correlation of 0.5 between successive daily values. Thus the correlations between input and output tracer fluctuations at lag zero (and thus the new water fractions) are artificially inflated by leakage from the stronger correlations at longer lags, where the TTD is much

stronger. Numerical experiments show that the bias in the new water fractions disappears when the short-lag serial correlation in the input tracers is removed, supporting this hypothesis for how the bias arises. Nonetheless, real-world precipitation tracer concentrations are often serially correlated, so researchers should be aware of the bias that they could potentially introduce into new water fractions if transit time distributions are humped. In the example shown here, this bias is effectively eliminated if the profiles of new water fractions are estimated from weekly samples instead of daily samples (see Fig. 10). This occurs because the input tracers are less correlated over weekly sampling intervals than over daily sampling intervals, and because the TTD is much stronger at short lags on weekly time scales (Fig. 8) than on daily time scales (Fig. 6). In real-world cases, biases like those shown in Fig. 9 may not be obvious, because the correct answer (shown here by the gray line, derived from benchmark model age tracking) will not be known. However, the behavior in Fig. 9b is implausible on hydrological grounds (why should catchments quickly transmit a particularly large fraction of very small precipitation events to the stream?), and the ${}^{\mathrm{P}}F_{\mathrm{new}}$ profiles in Figs. 9b and 10b show strongly contrasting patterns. Thus observations like these may help in identifying biased new water fraction estimates, even in cases where the TTD itself has not been quantified.

### 4.4 Visualizing catchment nonlinearity using precipitation- and discharge-filtered TTDs

Transit time distributions are typically constructed from the entire available tracer time series for a catchment, as in Figs. 5, 6, and 8. Such TTDs can be considered as averages of catchments' nonstationary transport behavior, as shown in Sect. 4.2 above. However, ensemble hydrograph separation can also be used to calculate TTDs for filtered subsets of the full catchment time series, focusing on either discharge or precipitation time steps that highlight particular conditions of interest. (In Appendix B we describe the new procedure that EHS_TTD uses for filtering precipitation time steps; this approach yields more accurate results than the one outlined in Sect. 4.2 of K2019.) TTDs from these filtered subsets of the full time series can yield further insights into catchment transport phenomena.

For example, we can map out the nonlinearities that give rise to catchments' nonstationary behavior, by comparing TTDs from subsets of the original time series that represent different catchment conditions (Fig. 11). Larger precipitation events in our benchmark model result in forward transit time distributions with peaks that are higher, earlier, and narrower (Fig. 11a). A similar progression in peak height, timing, and width is observed in forward TTDs (Fig. 11b) obtained from the benchmark tracer time series by setting the point filter $Pfilter$ in EHS_TTD to focus on individual ranges of precipitation rates. The backward transit time distributions in the benchmark model (Fig. 11c) differ somewhat from the forward transit time distributions (Fig. 11a), but exhibit a similar shift to higher, earlier, and narrower peaks at higher discharges. This trend is also reflected in backward TTDs (Fig. 11d) obtained from the benchmark time series by setting $Qfilter$ for the same discharge ranges used in Fig. 11c.

The ensemble hydrograph separation TTDs do not perfectly match the age tracking results shown by the dotted gray lines in Figs. 11b and 11d, particularly for the smallest fractions of the precipitation and discharge distributions, where fewer data points are available. Nonetheless, although the TTDs differ in detail from the age tracking results, they exhibit very similar progressions in peak height and shape, reflecting the nonlinearity in the benchmark model storages, which have shorter effective storage times at higher storage levels and discharges. Although the particular results shown in Fig. 11 are generated by a synthetic benchmark model, they illustrate how similar analyses could be used to infer nonlinear transport processes from real-world catchment data. Comparing TTDs representing different levels of antecedent catchment wetness, for example, could potentially be used to determine how much more precipitation bypasses catchment storage during wet conditions. Similarly, TTDs representing different levels of subsequent precipitation (over the following day or week, for example) could potentially be used to determine how effectively such precipitation mobilizes previously stored water. Thus Fig. 11 illustrates how TTDs from carefully selected subsets of catchment tracer time series can be used as fingerprints of catchment response, and as a basis for inferring the mechanisms underlying catchment behavior.

### 4.5 Choosing the number of TTD lags

An obvious question for users is the number of lags over which the TTD should be estimated. Here there is no fixed rule; the answer will depend on the time scales of interest, the length of the available tracer time series, and the shape of the TTD itself (which of course will not be known in advance). An empirical approach is to compare the results for several different maximum lags $m$, and see where the resulting TTDs are similar and different. Figure 12 shows this approach applied to daily and weekly tracer time series, yielding TTDs with contrasting shapes. The upper row (panels a and b) show L-shaped TTDs estimated from the same synthetic tracer time series that underlie Figs. 1-5, and the lower row shows humped TTDs from the same benchmark model driven by the same inputs, but with different parameters as described above. In each panel, the shorter and longer TTDs (shown in different colors) are generally consistent with one another, except in the case of the 4-lag TTD shown in blue in Fig. 12d. In that case, such a short TTD is evidently unable to capture the shape of the benchmark distribution, as indicated by its deviation from the TTDs of other lengths. One can also see that the last few lags of any TTD can diverge from the TTD shape defined by the other TTDs. In Fig. 12a the last few lags generally deviate downward and in Fig. 12c they generally deviate upward; thus there appears to be no general rule except that the last few lags of any TTD estimates should be treated with caution and potentially excluded from analysis.

A further observation from Fig. 12 is that TTD estimates from weekly tracer data may be at least as accurate, if not more so, than those calculated from daily tracer data. This may seem surprising, particularly because the time series underlying Fig. 12 are all five years long; thus the daily time series contain 7 times as many individual tracer measurements than the weekly time series. Nonetheless, for several reasons it is not surprising that in this case one could obtain more stable estimates from fewer data points. First of all, in these numerical experiments the precipitation tracer concentrations are serially correlated (as they also often are in the real world); thus there is more redundancy among the daily tracer inputs than among the weekly

Hydrology and Earth System Sciences
EGU
Discussions

tracer inputs. Secondly, the precipitation volumes are less variable (in percentage terms) from week to week than they are from day to day, meaning that the weekly calculations use fewer input concentrations that are accompanied by very small water volumes (and that therefore could not have much influence on the real-world catchment). And thirdly, lower sampling frequencies entail TTDs with coefficients at more widely spaced lags, which are thus less redundant with one another and thus can be individually constrained better. Of course with lower-frequency sampling one loses the short-lag tail of the TTD, which may be of particular interest. But in cases where this information is not crucial – or where only lower-frequency data are available – it appears that TTDs can be reliably estimated from samples taken weekly, and perhaps at even lower sampling frequencies.

## 5. Closing comments

In this short contribution, we have presented scripts that implement the ensemble hydrograph separation approach. We have also illustrated some of its quirks and limitations using synthetic data. The issues have been revealed through benchmark tests that are substantially stricter than many in the literature. One should not assume that other methods have fewer quirks and limitations, unless those methods have been tested with equal rigor.

For example, many benchmark data sets are generated using the same assumptions that underlie the analysis methods that they are used to test. Although the results of such tests often look nice, they are unrealistic because those idealized assumptions are unlikely to hold in real-world cases. For example, the TTD methods presented here would work very well if they were tested against benchmark data generated from a stationary TTD (see Fig. 7b), but this is hardly surprising since the regression in Eq. 2 assumes stationarity. But such a test is far removed from the real world, in which tracer data typically come from nonstationary catchment systems. Tests with nonstationary benchmarks yield results that are less (artificially) pleasing, but more realistic (e.g., Fig. 7a). These tests also demonstrate an important point, by showing how well the TTD method estimates the average of the time-varying TTDs that are likely to arise in real-world cases (see also Sect. 4 and Appendix B of K2019).

Although these scripts have been tested against several widely differing benchmark data sets (both here and in K2019), we encourage users to test them with their own benchmark data to verify that they are behaving as expected. As the examples presented here show, ensemble hydrograph separation can potentially be applied not only to the high-frequency tracer data sets that are now becoming available, but also to longer-term, lower-frequency tracer data that have been collected through many environmental monitoring programs. We hope that the availability of these scripts facilitates new and interesting explorations of the transport behavior of many different catchment systems.

**Author contributions**

J.W.K. wrote the R scripts and J.L.A.K. translated them into MATLAB. J.W.K. conducted the benchmark tests, drew the figures, and wrote the first draft of the manuscript. Both authors discussed the results and revised the manuscript.

**Data availability**

After acceptance of the manuscript, the R and MATLAB scripts and benchmark data sets will be available from a doi-referenced archive. For review purposes these files can be obtained from https://www.envidat.ch/#/metadata/ensemble-hydrograph-separation.

**Appendix A: Improved solution method for transit time distributions**

Ensemble hydrograph separation estimates transit time distributions by a multiple regression of streamflow tracer fluctuations against current and previous precipitation tracer fluctuations (Eq. 2, which is the counterpart to Eq. 1 over multiple lag intervals $k$). Performing this multiple regression with real-world data requires addressing the "missing data problem": precipitation tracer concentrations will be inherently unavailable during time steps where no precipitation falls, and both precipitation and streamflow tracer concentrations may also be missing due to sampling and measurement failures. The scripts presented here handle missing data somewhat differently than the procedure outlined in Sect. 4.2 of K2019. In this appendix we outline the new procedure and explain why it is necessary.

Equation (2) in the main text has the form of a multiple linear regression equation,

$$y_j = \sum_{k=0}^{m} \beta_k \, x_{j,k} + \alpha + \varepsilon_j \quad , \tag{A1}$$

where

$$y_j = C_{\mathrm{Q}_j} - C_{\mathrm{Q}_{j-m-1}} \tag{A2}$$

and

$$x_{j,k} = C_{\mathrm{P}_{j-k}} - C_{\mathrm{Q}_{j-m-1}} \quad . \tag{A3}$$

The conventional least-squares solution to such a multiple regression is usually expressed in matrix form as

$$\hat{\boldsymbol{\beta}} = (\mathbf{X}^{\mathrm{T}}\mathbf{X})^{-1}\, \mathbf{X}^{\mathrm{T}} \boldsymbol{Y} \quad , \tag{A4}$$

where $\widehat{\boldsymbol{\beta}}$ is the vector of the regression coefficients $\beta_k$, $\boldsymbol{Y}$ is the vector of the reference-corrected streamflow tracer concentrations $y_j = C_{\mathrm{Q}_j} - C_{\mathrm{Q}_{j-m-1}}$ and $\mathbf{X}$ is the matrix of the reference-corrected input tracer concentrations $x_{j,k} = C_{P_{j-k}} - C_{\mathrm{Q}_{j-m-1}}$ at each time step $j$ and lag $k$.

Equation (A4) cannot be applied straightforwardly to real-world catchment data, because it cannot be solved when values of $y_j$ or $x_{j,k}$ are missing. K2019 handled this missing data problem using a variant of Glasser's (1964) "pairwise deletion" approach. In this approach, Eq. (A4) was re-cast in terms of covariances,

$$\left(\hat{\beta}_{\mathrm{k}}\right) = \left( \quad \mathrm{cov}(\boldsymbol{X}_k, \boldsymbol{X}_\ell)_{(k\ell)} \quad \right)^{-1} \left( \mathrm{cov}(\boldsymbol{X}_k, \boldsymbol{Y})_{(ky)} \right) \quad , \tag{A5}$$

and these covariances were evaluated only for pairs of non-missing values (see Eqs. 42 and 43 of K2019), as signified by the subscripts in parentheses (i.e., they were only evaluated for time steps $j$ where both $x_{j,k}$ and $x_{j,\ell}$ or $x_{j,k}$ and $y_j$ were non-missing). The solution method presented in K2019 recognized that these covariances must be adjusted to account for the two different reasons that values can be missing. A value of $x_{j,k}$ that is missing because of a sampling or analysis failure represents missing information; it cannot be included in calculating the corresponding covariances, but (barring biases in which values are missing) it should have no systematic effect. By contrast, a value of $x_{j,k}$ that is missing because too little rain fell should dilute the covariances in which it appears, because with trivial precipitation inputs, the missing tracer concentration could not cause any meaningful co-varying change in streamflow tracer concentrations. These considerations required the elements of the covariance matrix in (A5) to be adjusted using prefactors called $n_{x_k}$ and $n_{x_k x_l}$ that accounted for the number of precipitation samples that were missing for the two different reasons outlined above (see Eqs. 44-45 and Appendix B of K2019).

Practical experience since the publication of K2019 has revealed at least three important limitations in the approach outlined above (and detailed in Sect. 4.2 and Appendix B of K2019). First, although this approach can work well if values of $y_j$ or $x_{j,k}$ are missing at random, in non-random cases it can lead to the covariances being estimated from inconsistent sets of values. For example, if $y_j$ is missing for some particular $j$, the corresponding values of $x_{j,k}$ will still be used in estimating the covariance matrix $\mathrm{cov}(\boldsymbol{X}_k, \boldsymbol{X}_\ell)_{(k\ell)}$. This is advantageous if the values are missing at random, because all the covariances will include as many data pairs as possible. However, the covariance estimates could become inconsistent, with potentially substantial consequences for the solution to Eq. (A5), if the values are missing non-randomly. In our case, the missing values are inherently structured, because a single missing precipitation tracer concentration $C_{\mathrm{P}_j}$ causes a diagonal line of missing values in $\mathbf{X}$, and a single missing streamflow tracer concentration $C_{\mathrm{Q}_j}$ causes a missing row in $\mathbf{X}$ and two missing values in $\boldsymbol{Y}$. The second problem is that our robust estimation procedure depends on iteratively reweighted least squares

(IRLS), which in turn requires us to calculate the regression residuals, which is impossible for any time step $j$ that is missing either $y_j$ or any of the $x_{j,k}$. The third problem is that estimating the uncertainties in the TTD requires the error variance, which again requires calculating the residuals. This last problem can be circumvented by using Glasser's error variance formula (Eq. 52 of K2019), but K2019 warns that this formula can yield implausible results, including negative error variance values (which are of course logically impossible).

Here, rather than removing the missing values and using Glasser's error variance formula, instead we fill in the missing values and calculate the residuals directly by inverting Eq. (A1), thus facilitating both robust estimation using IRLS, and direct calculation of the error variance for purposes of uncertainty estimation. The key to this approach is that we subtract the means from $\boldsymbol{Y}$ and from each column $\boldsymbol{X}_k$ of $\mathbf{X}$, (or subtract the weighted means in case of volume-weighting), and then we fill in the missing values with zeroes. Because each of the variables has already been "centered" to have a mean of zero, the in-filled values of zero will have no effect on the solution of Eq. (A1); in statistical terms, they will exert no leverage. This approach also has the advantage that the intercept $\alpha$ in Eq. (A1) becomes zero and drops out of the problem.

Broadly speaking, the solution proceeds similarly to Sect. 4.2-4.4 of K2019, with several important differences. One is that the covariance matrix now requires different prefactors than the $n_{x_k}$ and $n_{x_k x_l}$ used in K2019 to account for the two different types of missing data, because missing values will affect the covariances differently now that they are being in-filled with zeroes. In principle, a value of $x_{j,k}$ that is missing because of a sampling or analysis failure represents missing information, so it should not alter the covariances in which it appears. However, those covariances will be diluted when the missing value is replaced by zero (since it will add nothing to the cross-products in the numerator of the covariance formula, but will add to the total number $n$ in the denominator). The resulting covariances must be re-scaled to reverse this dilution artifact. By contrast, values of $x_{j,k}$ that are missing because no rain fell should dilute the covariances in which they appear, because with no precipitation, they could not cause any co-varying change in streamflow tracer concentrations. Thus replacing these missing values with zeroes correctly dilutes the corresponding covariances.

A sketch of the solution procedure is as follows. First we identify and remove outliers in the precipitation and streamflow tracer concentrations $C_\mathrm{P}$ and $C_\mathrm{Q}$ as described in Sects. 2.1 and 4.1, and use the remaining values to calculate the $y_j$ and $x_{j,k}$ using Eqs. (A2) and (A3). Next we calculate a matrix $u_{j,k}$ of boolean flags that indicate whether a given value of $x_{j,k}$ will be usable or not, according to the criteria outlined in the paragraph above: a value of $x_{j,k}$ is unusable if it is missing (and thus will need to be replaced by zero) and its corresponding value of precipitation is above p_threshold (and thus its contribution to the covariances would not be nearly zero anyway). If either of these conditions is not met, the value of $x_{j,k}$ is usable (potentially with replacement by zero if it is missing and corresponds to below-threshold precipitation inputs). In more explicit form,

$$u_{j,k} = \begin{cases} 0 & \text{if } x_{j,k} \text{ is missing and } P_{j-k} \geq P_{\text{threshold}} \\ 1 & \text{otherwise} \end{cases} \quad . \tag{A6}$$

We then eliminate any rows $j$ in $\mathbf{X}$, $\boldsymbol{Y}$, and $\mathbf{U}$ for which $y_j$ is missing and/or all of the $x_{j,k}$ are missing, because in such cases Eq. (A1) would have no meaningful solution. Next, we subtract the means (or the weighted means) from $\boldsymbol{Y}$ and from each column $\boldsymbol{X}_k$ of $\mathbf{X}$, and replace the missing values of $x_{j,k}$ with zeroes (there will be no missing values of $y_j$ at this stage).

If $robust$ = true, we then solve the multiple regression in Eq. (A1) using IRLS. We do not use the regression coefficients $\beta_k$ from the IRLS procedure, but instead use its robustness weights to down-weight anomalous points. These robustness weights are then combined with the volume weights (if $vol_wtd$ is true). The end result is a set of point weights $w_j$ that equal 1 times the volume weights (if any) times the IRLS robustness weights (if any), or that just equal 1 if $vol_wtd$ and $robust$ are both false.

We then calculate the (weighted) covariances as

$$\operatorname{cov}(\boldsymbol{X}_k, \boldsymbol{X}_\ell) = \frac{n_\mathrm{w}}{n_\mathrm{w}-1} \frac{\sum_j w_j \left(x_{j,k} - \bar{x}_k\right)\left(x_{j,\ell} - \bar{x}_\ell\right)}{\sum_j w_j} \tag{A7}$$

and

$$\operatorname{cov}(\boldsymbol{X}_k, \boldsymbol{Y}) = \frac{n_\mathrm{w}}{n_\mathrm{w}-1} \frac{\sum_j w_j \left(x_{j,k} - \bar{x}_k\right)\left(y_j - \bar{y}\right)}{\sum_j w_j} \quad , \tag{A8}$$

where $n_\mathrm{w}$ is the effective number of equally-weighted points,

$$n_\mathrm{w} = \frac{\left(\sum_j w_j\right)^2}{\sum_j \left(w_j^2\right)} \quad , \tag{A9}$$

which accounts for the unevenness of the weights $w_j$ (if all of the weights are equal, $n_\mathrm{w}$ equals $n$, the length of the vector $\boldsymbol{Y}$. The means shown in Eqs. (A7) and (A8) all equal zero, but they are preserved here so that the formulas can be readily recognized. To account for the contrasting types of missing values as outlined above, we multiply each of the covariances by prefactors $u_{x_k}/u_{x_k x_\ell}$, defined as

$$u_{x_k} = \sum_j w_j \, u_{j,k} \quad \text{and} \quad u_{x_k x_\ell} = \sum_j w_j \, u_{j,k} \, u_{j,\ell} \quad . \tag{A10}$$

With these prefactors, the solution to Eq. (A1) becomes

$$\left(\hat{\beta}_\mathrm{k}\right) = \left(u_{x_k}/u_{x_k x_\ell} \operatorname{cov}(\boldsymbol{X}_k, \boldsymbol{X}_\ell)\right)^{-1} \left(\operatorname{cov}(\boldsymbol{X}_k, \boldsymbol{Y})\right) \quad . \tag{A11}$$

To solve the same problem with Tikhonov-Phillips regularization, we instead solve

$$\left(\hat{\beta}_\mathrm{k}\right) = (\mathbf{C} + \lambda \mathbf{H})^{-1} \left(\operatorname{cov}(\boldsymbol{X}_k, \boldsymbol{Y})\right) \quad , \tag{A12}$$

where **C** is the covariance matrix $\left(u_{x_k}/u_{x_k x_\ell}\,\mathrm{cov}(\boldsymbol{X}_k,\boldsymbol{X}_\ell)\right)$, **H** is the Tikhonov-Phillips regularization matrix, and $\lambda$ controls the relative weight given to the regularization criterion (see Eqs. 49 and 50 of K2019).

To estimate the uncertainties in the regression coefficients $\hat{\beta}_{\mathrm{k}}$ we calculate the residuals by inverting Eq. (A1), recalling that $\alpha$=0,

$$\varepsilon_j = y_j - \sum_{k=0}^{m} \beta_k\, x_{j,k} \quad . \tag{A13}$$

We then calculate the (weighted) residual variance, accounting for both the degrees of freedom and any unevenness in the weights,

$$s_\varepsilon^2 = \frac{n-1}{n-(m+1)-1}\,\frac{n_{\mathrm{w}}}{n_{\mathrm{w}}-1}\,\frac{\sum_j w_j\left(\varepsilon_j-\bar{\varepsilon}\right)^2}{\sum_j w_j} \quad , \tag{A14}$$

where the (weighted) mean of the residuals should be zero, but it is included here for completeness. We then calculate the standard errors of the regression coefficients, following Eq. (54) of K2019, using

$$\mathrm{SE}(\hat{\beta}_k) = \sqrt{\frac{s_\varepsilon^2}{n_{\mathrm{eff}_k}}}\;\sqrt{\left[(\mathbf{C}+\lambda\mathbf{H})^{-1}(\mathbf{C})\quad(\mathbf{C}+\lambda\mathbf{H})^{-1}\right]_{kk}} \quad , \tag{A15}$$

but with the difference that the unevenness in the weighting is already taken into account in Eq. (A14), so the effective sample size is now calculated following Eq. (13) of K2019 as

$$n_{\mathrm{eff}_k} = \sum_j u_{j,k}\,\frac{1-r_{\mathrm{sc}}}{1+r_{\mathrm{sc}}} \quad , \tag{A16}$$

where $r_{sc}$ is the lag-1 (weighted) serial correlation in the residuals $\varepsilon_j$. If the *ser_corr* option is set to false, the effective sample size is calculated as

$$n_{\mathrm{eff}_k} = \sum_j u_{j,k} \quad . \tag{A17}$$

The regression coefficients and their standard errors are then converted into TTDs and their associated standard errors using Eqs. (55), (60), and (63)-(66) of K2019. Readers should note, however, that these scripts do not explicitly take account of the sampling interval and its units. Thus their results should be interpreted as being in reciprocal units of the sampling interval, e.g., day$^{-1}$ for daily sampling and week$^{-1}$ for weekly sampling.

**Appendix B: Improved method for filtering precipitation time steps in TTD estimation**

The system of equations that is used to estimate transit time distributions in ensemble hydrograph separation (Eq. A1) can be represented as a matrix equation of the form

$$\begin{pmatrix} y_1 \\ y_2 \\ y_3 \\ y_4 \\ y_5 \\ y_6 \\ \vdots \\ y_n \end{pmatrix} = \begin{pmatrix} x_{1,0} & x_{1,1} & x_{1,2} & \cdots & x_{1,m} \\ x_{2,0} & x_{2,1} & x_{2,2} & \cdots & x_{2,m} \\ x_{3,0} & x_{3,1} & x_{3,2} & \cdots & x_{3,m} \\ x_{4,0} & x_{4,1} & x_{4,2} & \cdots & x_{4,m} \\ x_{5,0} & x_{5,1} & x_{5,2} & \cdots & x_{5,m} \\ x_{6,0} & x_{6,1} & x_{6,2} & \cdots & x_{6,m} \\ \vdots & \vdots & \vdots & \ddots & \vdots \\ x_{n,0} & x_{n,1} & x_{n,2} & \cdots & x_{n,m} \end{pmatrix} \begin{pmatrix} \beta_0 \\ \beta_1 \\ \beta_2 \\ \vdots \\ \beta_m \end{pmatrix} + \begin{pmatrix} \alpha \\ \alpha \\ \alpha \\ \alpha \\ \alpha \\ \alpha \\ \vdots \\ \alpha \end{pmatrix} + \begin{pmatrix} \varepsilon_1 \\ \varepsilon_2 \\ \varepsilon_3 \\ \varepsilon_4 \\ \varepsilon_5 \\ \varepsilon_6 \\ \vdots \\ \varepsilon_n \end{pmatrix} , \tag{B1}$$

where $x_{j,k}$ expresses the concentration of the tracer input which enters the catchment at time step $i = j - k$, at a lag of $k$ time steps before some of it leaves the catchment at time step $j$ as part of the discharge concentration $y_j$, with both concentrations normalized as described in Eqs. (A2) and (A3). Filtering this system of equations according to discharge time steps (so that, for example, periods with low discharge are excluded) is accomplished straightforwardly by deleting the corresponding rows from the matrices. For example, if we want to exclude discharge time steps 3, 4, and 5, Eq. (B1) becomes

$$\begin{pmatrix} y_1 \\ y_2 \\ y_6 \\ \vdots \\ y_n \end{pmatrix} = \begin{pmatrix} x_{1,0} & x_{1,1} & x_{1,2} & \cdots & x_{1,m} \\ x_{2,0} & x_{2,1} & x_{2,2} & \cdots & x_{2,m} \\ x_{6,0} & x_{6,1} & x_{6,2} & \cdots & x_{6,m} \\ \vdots & \vdots & \vdots & \ddots & \vdots \\ x_{n,0} & x_{n,1} & x_{n,2} & \cdots & x_{n,m} \end{pmatrix} \begin{pmatrix} \beta_0 \\ \beta_1 \\ \beta_2 \\ \vdots \\ \beta_m \end{pmatrix} + \begin{pmatrix} \alpha \\ \alpha \\ \alpha \\ \vdots \\ \alpha \end{pmatrix} + \begin{pmatrix} \varepsilon_1 \\ \varepsilon_2 \\ \varepsilon_6 \\ \vdots \\ \varepsilon_n \end{pmatrix} , \tag{B2}$$

which can be solved in exactly the same way as the full system of equations shown in Eq. (B1). Filtering according to precipitation time steps (so that, for example, periods dry antecedent conditions are excluded) is less straightforward. The approach outlined in Sect. 4.8 of K2019 is to simply exclude the corresponding values of $x_{j,k}$, which form diagonal stripes in the **X** matrix. For example, for an artificially simplified case with only nine discharge time steps and four lags, these diagonal stripes of missing values would appear as

$$\begin{pmatrix} y_1 \\ y_2 \\ y_3 \\ y_4 \\ y_5 \\ y_6 \\ y_7 \\ y_8 \\ y_9 \end{pmatrix} = \begin{pmatrix} - & - & x_{1,2} & x_{1,3} \\ x_{2,0} & - & - & x_{2,3} \\ x_{3,0} & x_{3,1} & - & - \\ - & x_{4,1} & x_{4,2} & - \\ x_{5,0} & - & x_{5,2} & x_{5,3} \\ - & x_{6,1} & - & x_{6,3} \\ x_{7,0} & - & x_{7,2} & - \\ x_{8,0} & x_{8,1} & - & x_{8,3} \\ x_{9,0} & x_{9,1} & x_{9,2} & - \end{pmatrix} \begin{pmatrix} \beta_0 \\ \beta_1 \\ \beta_2 \\ \beta_3 \end{pmatrix} + \begin{pmatrix} \alpha \\ \alpha \\ \alpha \\ \alpha \\ \alpha \\ \alpha \\ \alpha \\ \alpha \\ \alpha \end{pmatrix} + \begin{pmatrix} \varepsilon_1 \\ \varepsilon_2 \\ \varepsilon_3 \\ \varepsilon_4 \\ \varepsilon_5 \\ \varepsilon_6 \\ \varepsilon_7 \\ \varepsilon_8 \\ \varepsilon_9 \end{pmatrix} . \tag{B3}$$

The technical problem of performing such a calculation can be solved as described in Appendix A above, but this alone will not solve the *mathematical* problem created by the diagonal stripes of missing values. The mathematical problem is that the

influence of the missing $x_{j,k}$ will still be reflected in the fluctuations in the discharge tracer concentrations $y_j$, and any regression solution will seek to explain those fluctuations in terms of the $x_{j,k}$ values that remain, thus biasing the regression coefficients $\beta_k$. A better approach than Eq. (B3) is not to remove the excluded values entirely, but instead to separate them in a new group of variables with their own coefficients, as follows:

$$\begin{pmatrix} y_1 \\ y_2 \\ y_3 \\ y_4 \\ y_5 \\ y_6 \\ y_7 \\ y_8 \\ y_9 \end{pmatrix} = \begin{pmatrix} - & - & x_{1,2} & x_{1,3} \\ x_{2,0} & - & - & x_{2,3} \\ x_{3,0} & x_{3,1} & - & - \\ - & x_{4,1} & x_{4,2} & - \\ x_{5,0} & - & x_{5,2} & x_{5,3} \\ - & x_{6,1} & - & x_{6,3} \\ x_{7,0} & - & x_{7,2} & - \\ x_{8,0} & x_{8,1} & - & x_{8,3} \\ x_{9,0} & x_{9,1} & x_{9,2} & - \end{pmatrix} \begin{pmatrix} \beta_0 \\ \beta_1 \\ \beta_2 \\ \beta_3 \end{pmatrix} + \begin{pmatrix} x'_{1,0} & x'_{1,1} & - & - \\ - & x'_{2,1} & x'_{2,2,} & - \\ - & - & x'_{3,2} & x'_{3,3} \\ x'_{4,0} & - & - & x'_{4,2} \\ - & x'_{5,1} & - & - \\ x'_{6,0} & - & x'_{6,2} & - \\ - & x'_{7,1} & - & x'_{7,3} \\ - & - & x'_{8,2} & - \\ - & - & - & x'_{9,2} \end{pmatrix} \begin{pmatrix} \beta'_0 \\ \beta'_1 \\ \beta'_2 \\ \beta'_3 \end{pmatrix} + \begin{pmatrix} \alpha \\ \alpha \\ \alpha \\ \alpha \\ \alpha \\ \alpha \\ \alpha \\ \alpha \\ \alpha \end{pmatrix} + \begin{pmatrix} \varepsilon_1 \\ \varepsilon_2 \\ \varepsilon_3 \\ \varepsilon_4 \\ \varepsilon_5 \\ \varepsilon_6 \\ \varepsilon_7 \\ \varepsilon_8 \\ \varepsilon_9 \end{pmatrix} . \tag{B4}$$

In Eq. (B4), each of the original input tracer values $x_{j,k}$ either appears in the left-hand matrix of included values (denoted $x_{j,k}$) and is multiplied by the corresponding coefficient $\beta_k$, or, if it has been filtered out, appears in the right-hand matrix of excluded values (denoted $x'_{j,k}$) and is multiplied by the corresponding coefficient $\beta'_k$. Equation (B4) suppresses the distortion of the $\beta_k$ coefficients from the missing $x_{j,k}$, because each row of this matrix equation retains all the values in the original equation (Eq. B2), but now has separate sets of coefficients for the included values and the excluded values. We can merge these two sets of coefficients and combine the **X** and **X′** matrices, re-casting Eq. (B4) as a conventional regression problem,

$$\begin{pmatrix} y_1 \\ y_2 \\ y_3 \\ y_4 \\ y_5 \\ y_6 \\ y_7 \\ y_8 \\ y_9 \end{pmatrix} = \begin{pmatrix} - & - & x_{1,2} & x_{1,3} & x'_{1,0} & x'_{1,1} & - & - \\ x_{2,0} & - & - & x_{2,3} & - & x'_{2,1} & x'_{2,2,} & - \\ x_{3,0} & x_{3,1} & - & - & - & - & x'_{3,2} & x'_{3,3} \\ - & x_{4,1} & x_{4,2} & - & x'_{4,0} & - & - & x'_{4,2} \\ x_{5,0} & - & x_{5,2} & x_{5,3} & - & x'_{5,1} & - & - \\ - & x_{6,1} & - & x_{6,3} & x'_{6,0} & - & x'_{6,2} & - \\ x_{7,0} & - & x_{7,2} & - & - & x'_{7,1} & - & x'_{7,3} \\ x_{8,0} & x_{8,1} & - & x_{8,3} & - & - & x'_{8,2} & - \\ x_{9,0} & x_{9,1} & x_{9,2} & - & - & - & - & x'_{9,3} \end{pmatrix} \begin{pmatrix} \beta_0 \\ \beta_1 \\ \beta_2 \\ \beta_3 \\ \beta'_0 \\ \beta'_1 \\ \beta'_2 \\ \beta'_3 \end{pmatrix} + \begin{pmatrix} \alpha \\ \alpha \\ \alpha \\ \alpha \\ \alpha \\ \alpha \\ \alpha \\ \alpha \\ \alpha \end{pmatrix} + \begin{pmatrix} \varepsilon_1 \\ \varepsilon_2 \\ \varepsilon_3 \\ \varepsilon_4 \\ \varepsilon_5 \\ \varepsilon_6 \\ \varepsilon_7 \\ \varepsilon_8 \\ \varepsilon_9 \end{pmatrix} , \tag{B5}$$

which can be solved by the approach outlined in Appendix A. One important detail is that the Tikhonov-Phillips regularization matrix must be segmented so that regularization is applied separately to the $\beta_k$ and the $\beta'_k$; otherwise a regularization algorithm would try to smooth over the jump between $\beta_m$, which will typically be small, and $\beta'_0$, which could be large. Regularization can be applied separately to the two sets of coefficients by configuring the regularization matrix as

$$\begin{pmatrix} \begin{pmatrix} \mathbf{H} \end{pmatrix} & \begin{pmatrix} \mathbf{0} \end{pmatrix} \\ \begin{pmatrix} \mathbf{0} \end{pmatrix} & \begin{pmatrix} \mathbf{H} \end{pmatrix} \end{pmatrix} , \qquad \text{(B6)}$$

where the diagonal sub-matrices **H** are $m$-by-$m$ Tikhonov-Phillips regularization matrices (see Eq. 49 of K2019), and the off-diagonal sub-matrices are $m$-by-$m$ matrices of zeroes.

Benchmark tests verify that the approach outlined in Eq. (B5) yields much more accurate estimates of $\beta_k$ than the approach outlined in K2019 does. Therefore this approach is employed in EHS_TTD whenever the input data are filtered according to precipitation time steps.

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

# Figures

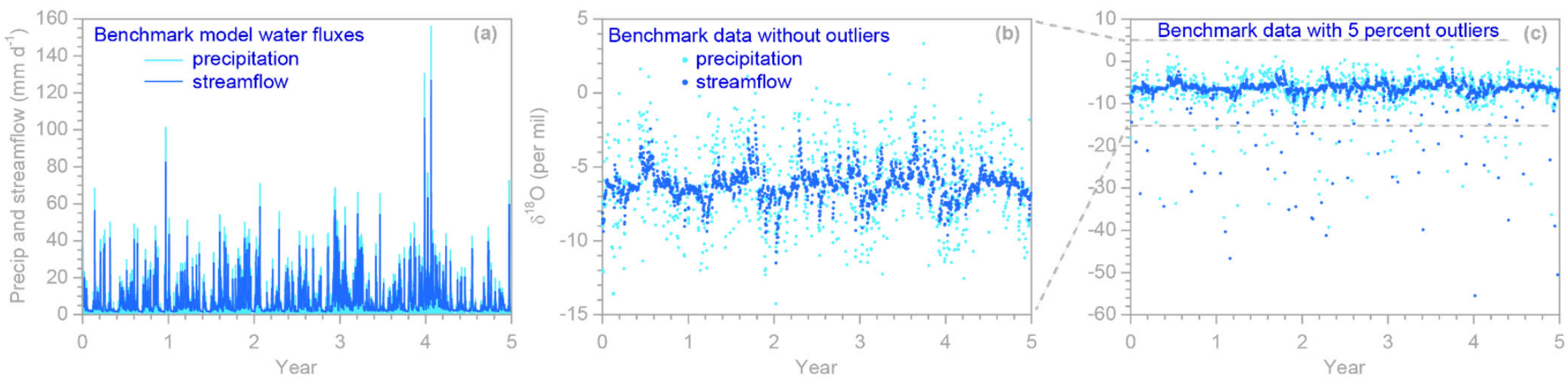

**Figure 1.** Benchmark model daily water fluxes (a) and precipitation and streamflow isotope time series (light and dark blue symbols, respectively), without outliers (b) and with 5% outliers (c). The axis frame of the outlier-free data (panel b) corresponds to the dashed lines in panel (c). Benchmark model parameters are $S_{u,ref}$ = 50 mm, $S_{l,ref}$ = 2000 mm, $b_u$ = 10, $b_l$ = 3, and $\eta$ = 0.8. The model is driven by a daily precipitation time series from Plynlimon, Wales, and a hypothetical precipitation $\delta^{18}O$ isotope record with a seasonal sinusoidal amplitude of 1.2 per mil, a normally distributed random standard deviation of 2.5 per mil, and a serial correlation of 0.5 between successive daily isotope values. A random measurement error with a standard deviation of 0.1 per mil was added to all simulated precipitation and streamflow isotope measurements.

Hydrology and Earth System Sciences
EGU
Discussions

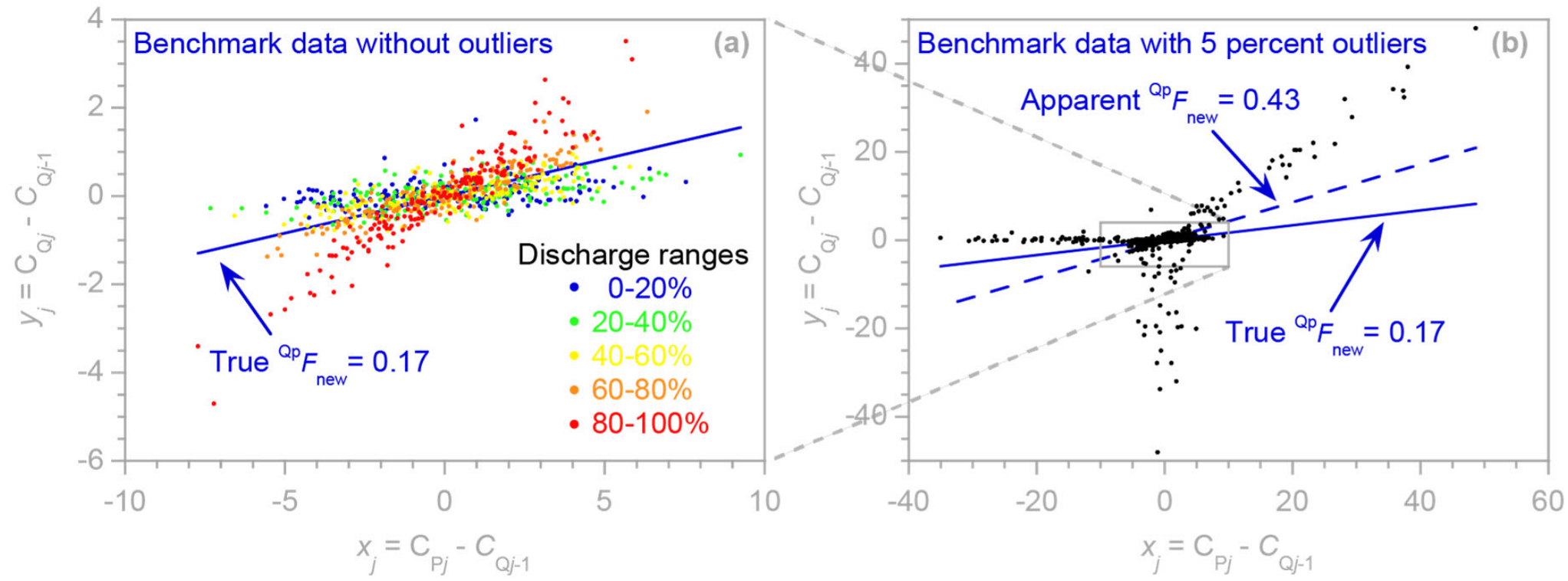

**Figure 2.** Regression relationship (Eq. 1) used to estimate the event new water fraction $^{Qp}F_{new}$, using (a) the outlier-free benchmark data of Fig. 1a, and (b) the outlier-corrupted benchmark data of Fig. 1b. The axis frame of the outlier-free plot (panel a) corresponds to the gray rectangle in panel (b). In panel (a), five different percentile ranges of the discharge distribution are shown in contrasting colors. The stronger coupling between tracer fluctuations in precipitation and streamflow at higher discharges reflects a larger new water contribution to streamflow. The event new water fraction $^{Qp}F_{new}$ is the average fraction of streamflow that is composed of precipitation that fell during the current sampling interval, and is calculated from the regression slope between fluctuations in precipitation and streamflow tracer concentrations ($C_{P_j}$ and $C_{Q_j}$), each expressed relative to the previous streamflow sample's tracer concentration ($C_{Q_{j-1}}$). Because this reference value appears on both axes of the regression plot, anomalous streamflow tracer values will appear as positively correlated outliers. These points, as well as the $x_j$ outliers generated by anomalous precipitation tracer values, may have substantial leverage on the fitted regression line, leading to distorted estimates of the regression slope $^{Qp}F_{new}$.

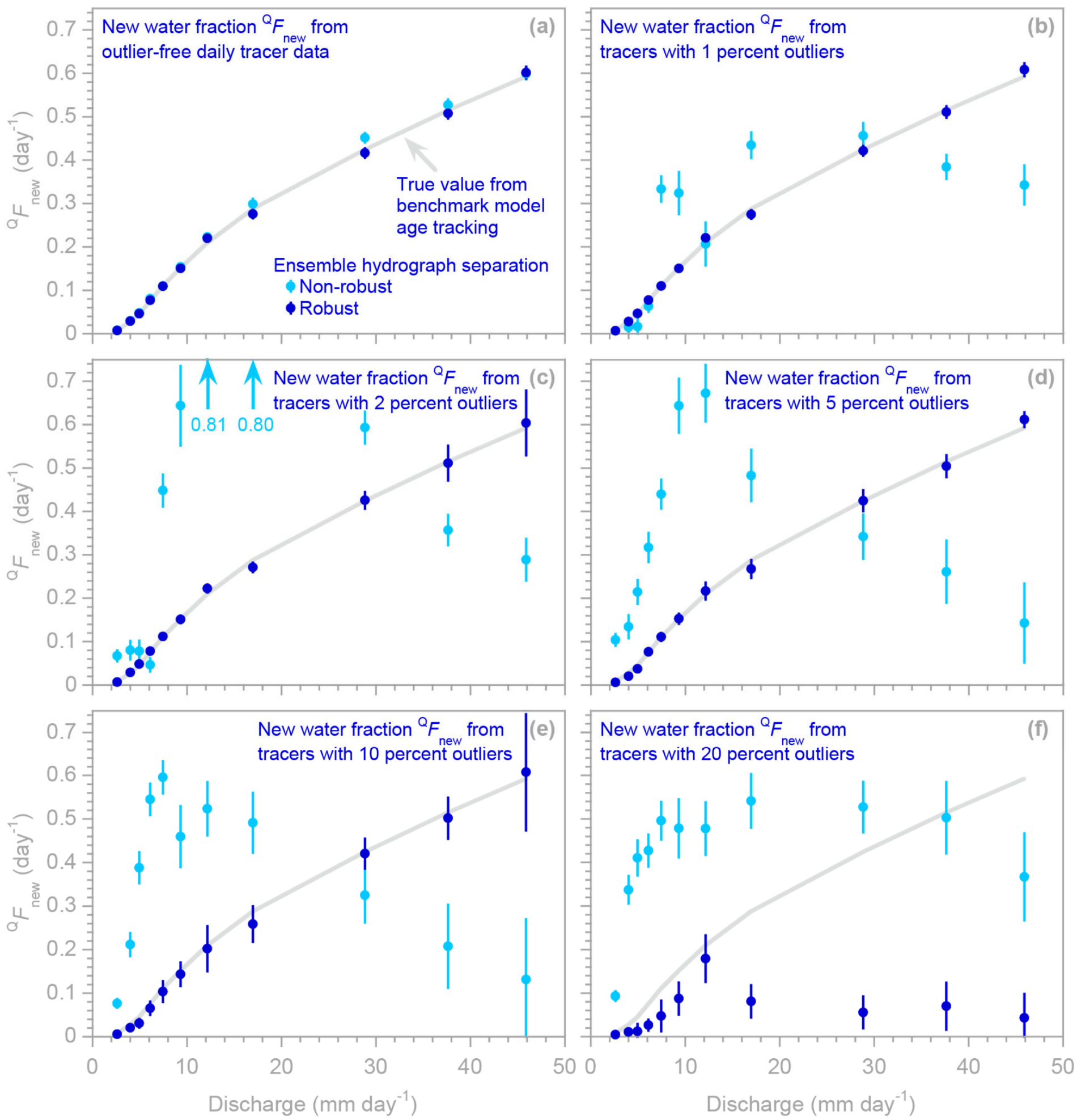

**Figure 3.** Profiles illustrating how new water fractions of discharge change with discharge regime, estimated using robust and non-robust methods (dark and light blue symbols, respectively; error bars indicate one standard error) applied to synthetic benchmark tracer data without different percentages of outliers. In profiles generated from outlier-free data (a), many light blue symbols are invisible because they are directly overprinted by dark blue symbols. The light gray line indicates the true new water fraction, as calculated from water age tracking in the benchmark model. The non-robust estimates (light blue symbols) closely follow the true values (gray line) if the tracer time series are outlier-free (a), but deviate markedly if they are corrupted by outliers (b-f), even if those outliers comprise only a few percent of the data (b-d). The robust estimates (dark blue symbols) closely follow the true values (gray line), until the outliers become so frequent that the robust estimation algorithm is no longer effective against them (f).

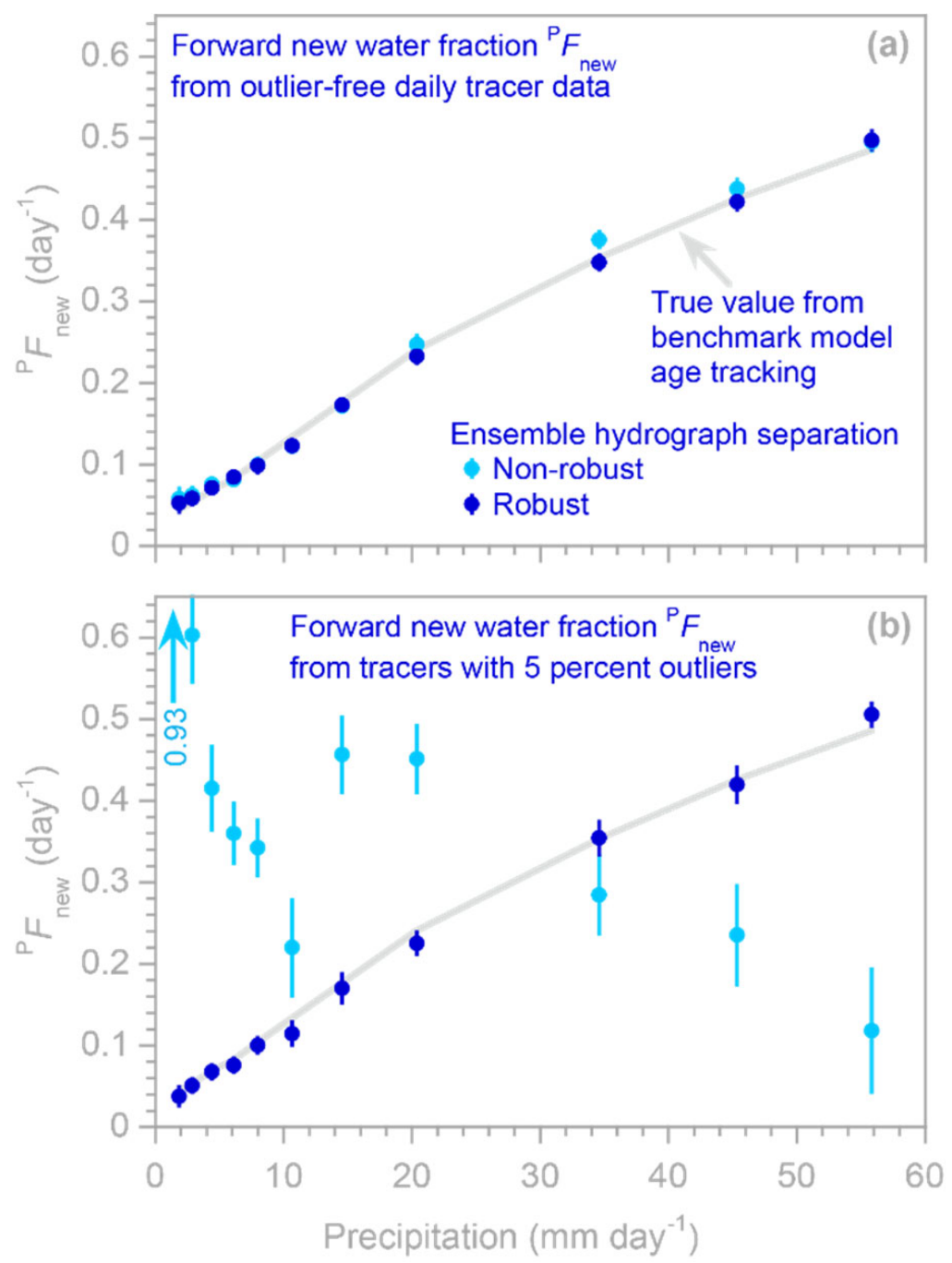

**Figure 4.** Profiles illustrating how "forward" new water fractions (new water fractions of precipitation, i.e., fractions of precipitation leaving as streamflow during the current sampling interval) change with precipitation regime, estimated using robust and non-robust methods (dark and light blue symbols, respectively; error bars indicate one standard error) applied to synthetic benchmark tracer data without outliers (a) and with outliers (b). In (a), many light blue symbols are invisible because they are directly overprinted by dark blue symbols. The light gray line indicates the true forward new water fraction, as calculated from water age tracking in the benchmark model. The robust estimates (dark blue symbols) closely follow this line, whether or not the benchmark data contain outliers. The non-robust estimates (light blue symbols) closely follow the gray line if the tracer time series are outlier-free (a), but deviate markedly if the tracer data are corrupted by outliers (b). One off-scale value is indicated by the light blue arrow.

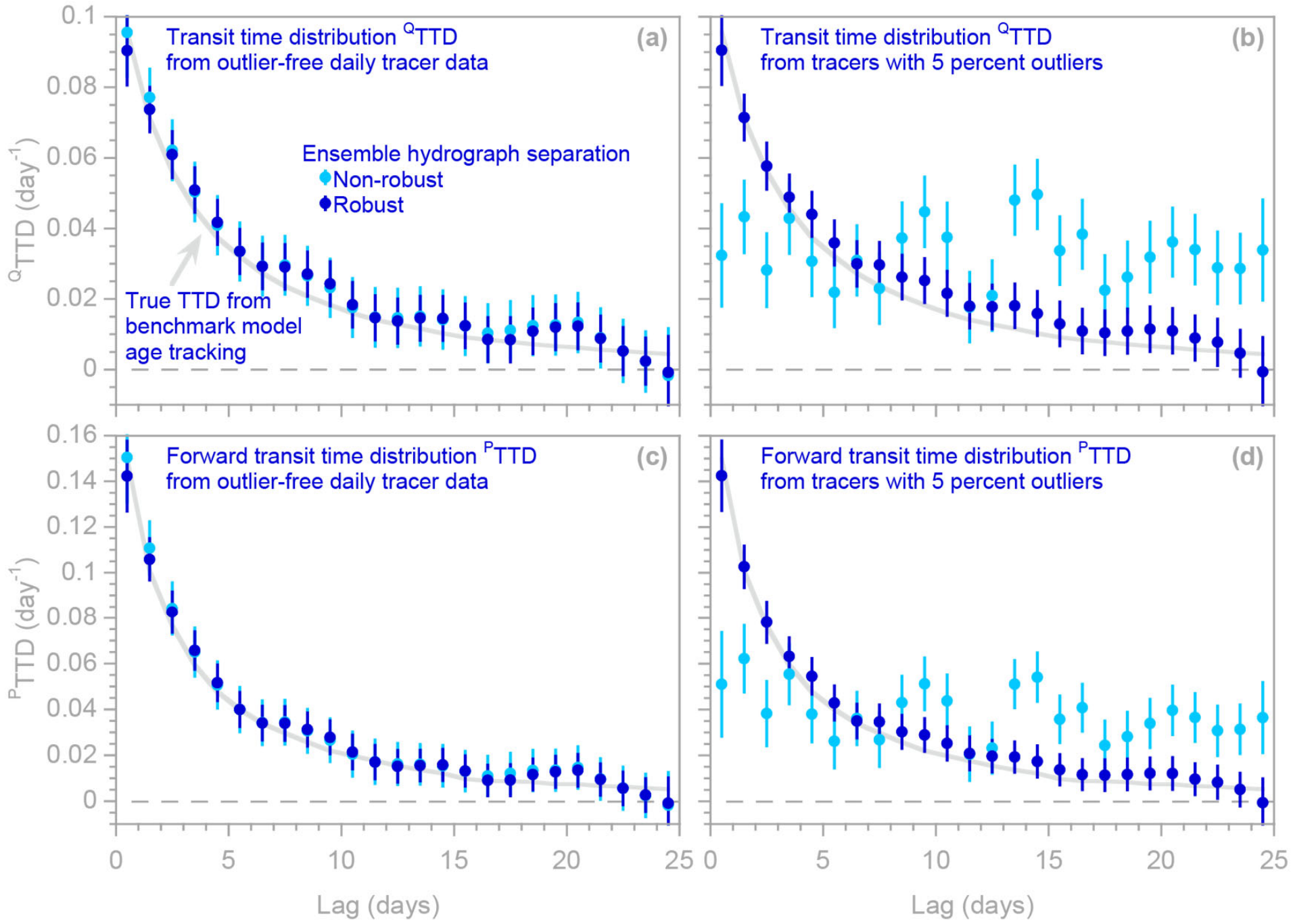

**Figure 5.** Transit time distributions of discharge ($^{Q}$TTDs; panels a and b) and forward new water fractions of precipitation ($^{P}$TTDs; panels c and d), as estimated by ensemble hydrograph separation from synthetic benchmark tracer time series without outliers (a, c) and with outliers (b, d). Symbols show results obtained with and without robust estimation (dark and light symbols, respectively); error bars indicate one standard error. In (a) and (c), many light blue symbols are obscured because they are overprinted by dark blue symbols. Light gray curves indicate the true TTDs, as calculated from water age tracking in the benchmark model. When the tracer data are free of outliers (a, c), estimates obtained from robust and non-robust methods are almost equally good, both typically lying within 1 standard error of the true TTDs (gray curves). However, when the input data are corrupted by extreme outliers (b, d), the non-robust method yields estimates (light symbols) that deviate substantially from the true TTDs, whereas the robust method (dark symbols) still follows the gray curve nearly as well as it did with the outlier-free data.

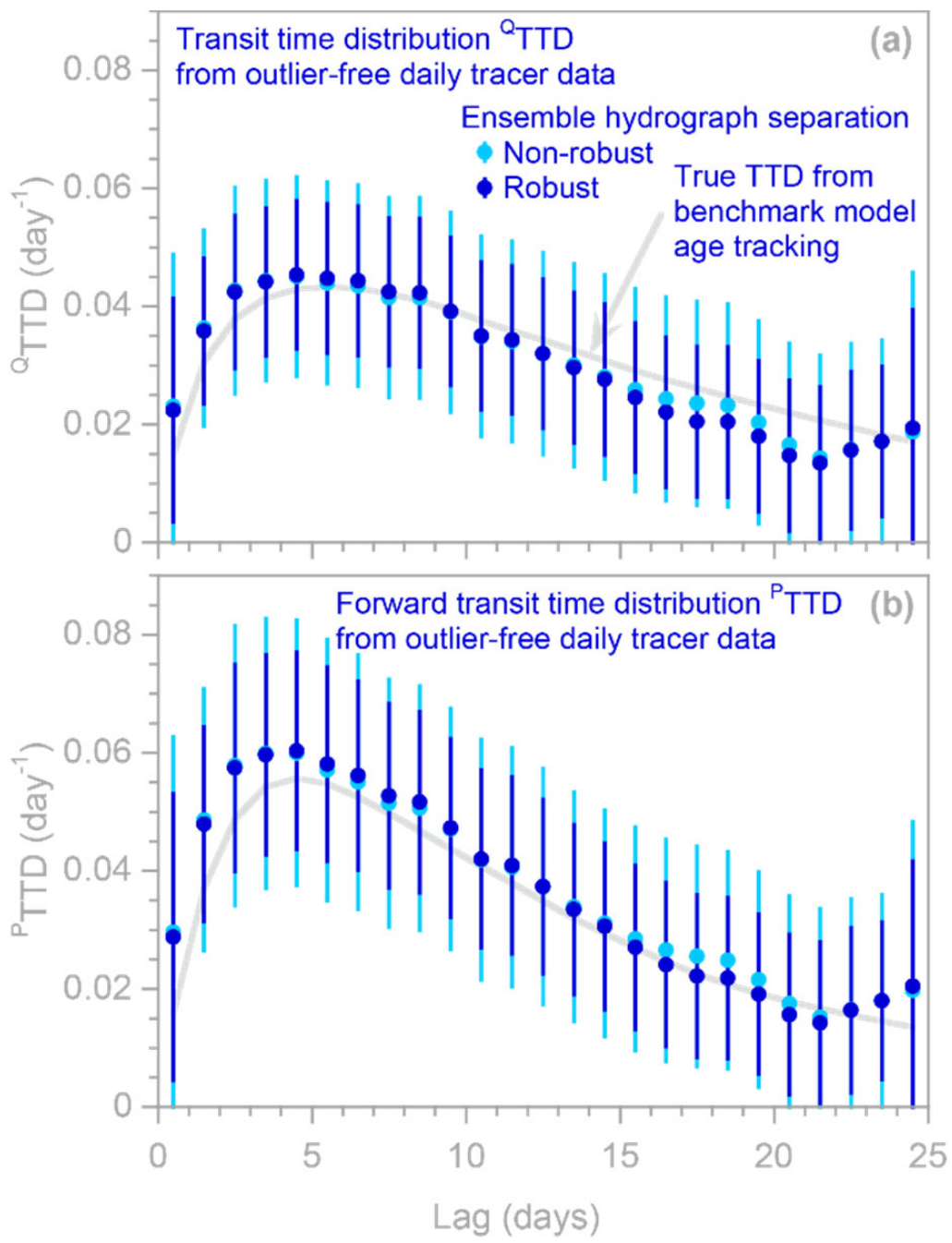

**Figure 6.** Humped transit time distributions ($^{Q}$TTDs) and forward transit time distributions (transit time distributions of precipitation, $^{P}$TTDs), as estimated by ensemble hydrograph separation from synthetic benchmark tracer time series. Symbols show ensemble hydrograph separation results obtained with and without robust estimation (dark and light symbols, respectively); error bars indicate one standard error. Many light blue symbols are obscured because they are overprinted by dark blue symbols. The light gray lines indicate the true TTD, as calculated from water age tracking in the benchmark model. The TTDs calculated from the synthetic tracer time series follow these gray lines, but the error bars indicate that the standard errors are overestimated by large factors. Benchmark model parameters are $S_{u,ref}$ = 50 mm, $S_{l,ref}$ = 50 mm, $b_u$ = 5, $b_l$ = 2, and $\eta$ = 0.01. The model is driven by the same time series of precipitation rates and $\delta^{18}O$ values as shown in Fig. 1 and as used in Figs. 2-5.

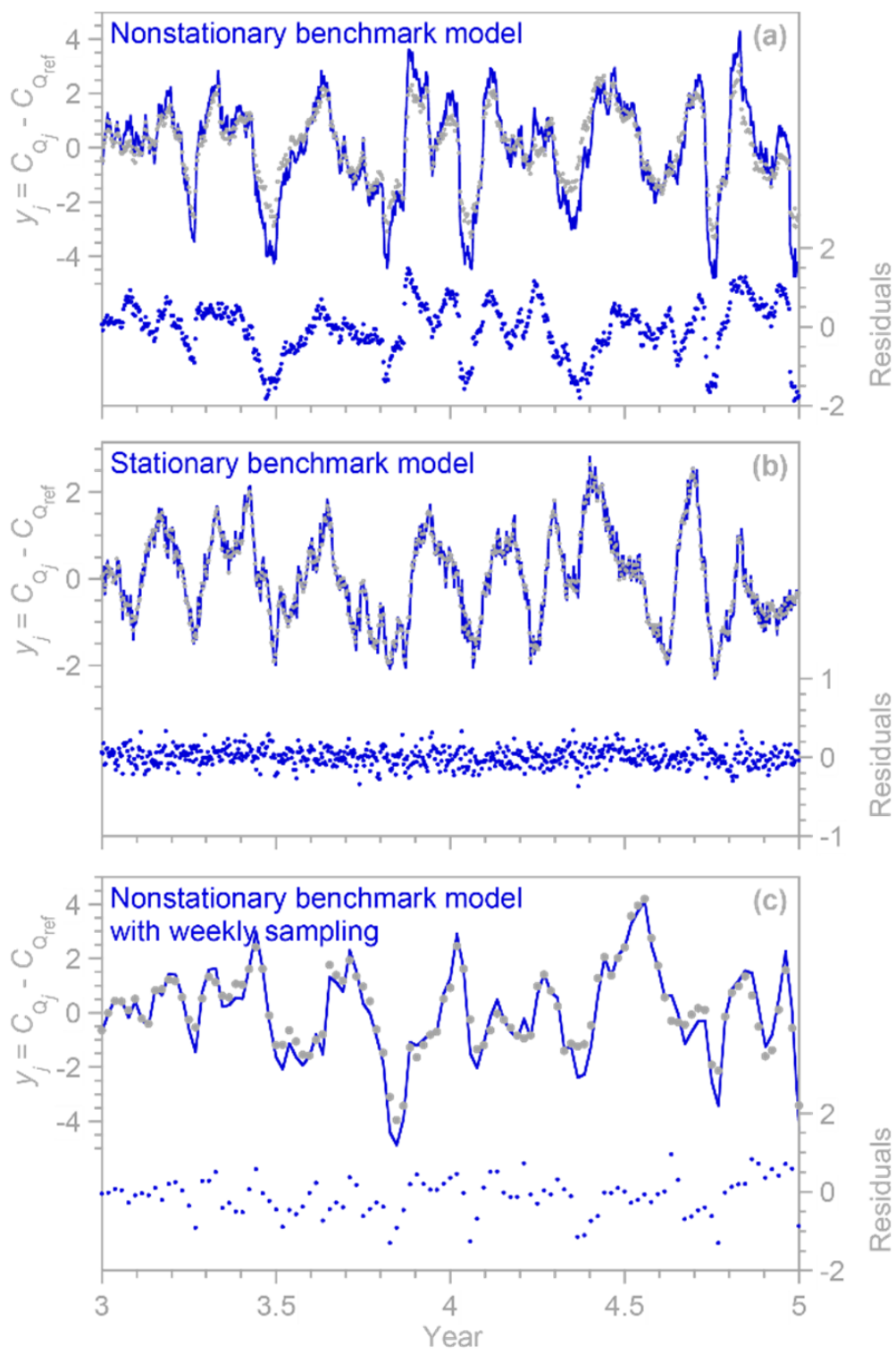

**Figure 7.** Comparison of observed and fitted streamflow tracer time series (gray dots and dark blue line, respectively, shown relative to their lagged reference values as in the left hand side of Eq. 2), and fitting residuals (dark blue dots), for the nonstationary benchmark model with a humped time-averaged TTD (a), for the same model with the same parameters, but with constant precipitation rates and therefore a stationary humped TTD (b), and for the same model based on weekly rather than daily sampling (c). The observed and fitted tracer time series are shown relative to the reference tracer concentration (the streamflow concentration beyond the longest TTD lag; see Kirchner, 2019 for details). In (a), the multiple regression fit to the streamflow tracers generally exhibits the correct behavior, but with minor errors in amplitude and timing, resulting in residuals that exhibit strong serial correlation (lag-1 $r_{sc}$ = 0.96) and thus greatly exaggerated standard errors of the regression coefficients that define the TTD. By contrast, under a stationary benchmark model (b), achieved by holding the precipitation rate constant at its average value, the multiple regression fit to the streamflow tracers yields much smaller residuals (note the difference in scale) with little serial correlation (lag-1 $r_{sc}$ = 0.23). Weekly samples from the nonstationary benchmark model (c) yield residuals with much less serial correlation (lag-1 $r_{sc}$ = 0.66) than daily samples (a), resulting in less exaggerated standard errors of the regression coefficients (compare Figs. 6 and 8).
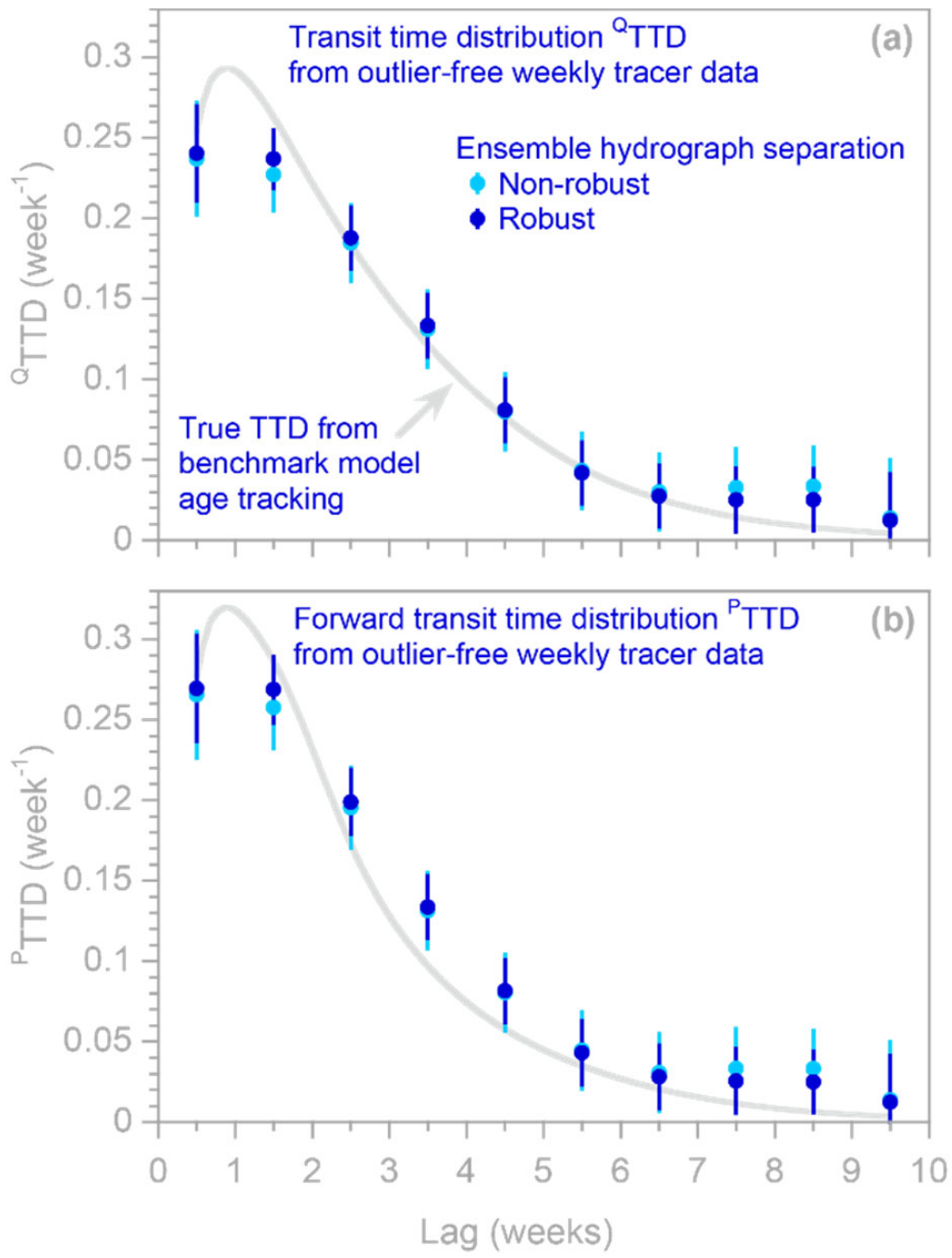

**Figure 8.** Transit time distributions ($^{Q}$TTDs) and forward transit time distributions (transit time distributions of precipitation, $^{P}$TTDs), as estimated by ensemble hydrograph separation from synthetic weekly benchmark tracer time series. Symbols show ensemble hydrograph separation results obtained with and without robust estimation (dark and light symbols, respectively); error bars indicate one standard error. Many light blue symbols are obscured because they are overprinted by dark blue symbols. The light gray lines indicates the true TTD, as calculated from water age tracking in the benchmark model. The tracer data used here are the same as in Fig. 6, but aggregated to simulate weekly instead of daily sampling. The standard errors are not as overestimated as those in Fig. 6, because weekly sampling results in weaker serial correlation in the residuals of the regressions that estimate the TTD (see Fig. 7).

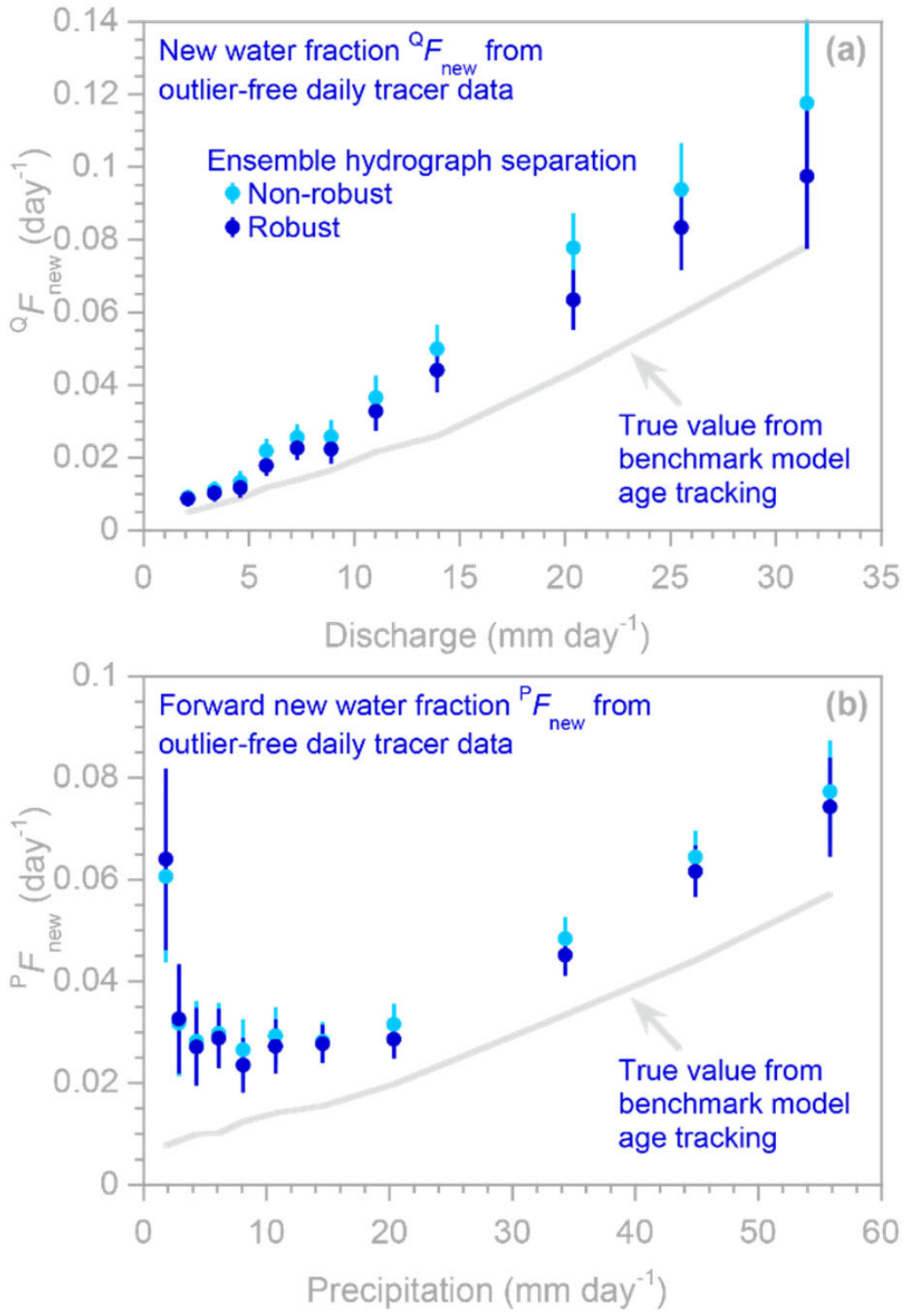

**Figure 9.** Profiles of new water fractions ( $^{Q}F_{\mathrm{new}}$, panel a) and forward new water fractions ( $^{P}F_{\mathrm{new}}$, panel b), estimated using robust and non-robust methods (dark and light blue symbols, respectively; error bars indicate one standard error) applied to daily tracer time series, generated by the benchmark model using parameters that generate a humped distribution (see Fig. 6). Some light blue symbols are invisible where they are overprinted by dark blue symbols. The light gray lines show the true new water fractions, as calculated from water age tracking in the benchmark model. These new water fractions are overestimated by both robust and non-robust methods.

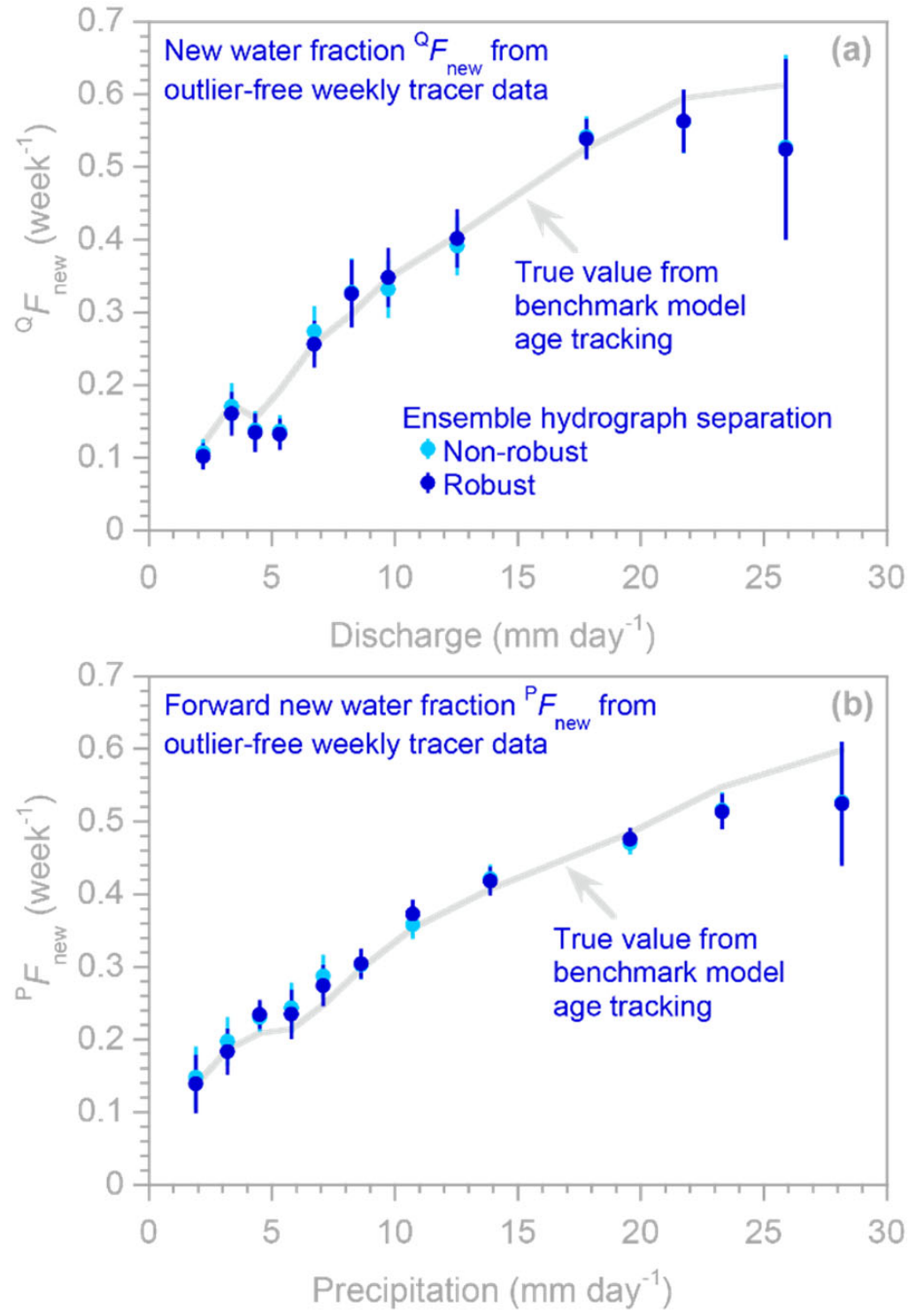

**Figure 10.** Profiles of new water fractions ($^{Q}F_{new}$, panel a) and forward new water fractions ($^{P}F_{new}$, panel b), estimated using robust and non-robust methods (dark and light blue symbols, respectively; error bars indicate one standard error) applied to weekly tracer time series from the benchmark model using parameters that generate a humped distribution (see Fig. 6). Some light blue symbols are invisible where they are overprinted by dark blue symbols. The light gray lines show the true new water fractions, as calculated from water age tracking in the benchmark model. In contrast to the results from daily time series (Fig. 9), weekly tracer time series yield new water fraction profiles that are consistent with the true values determined by age tracking.
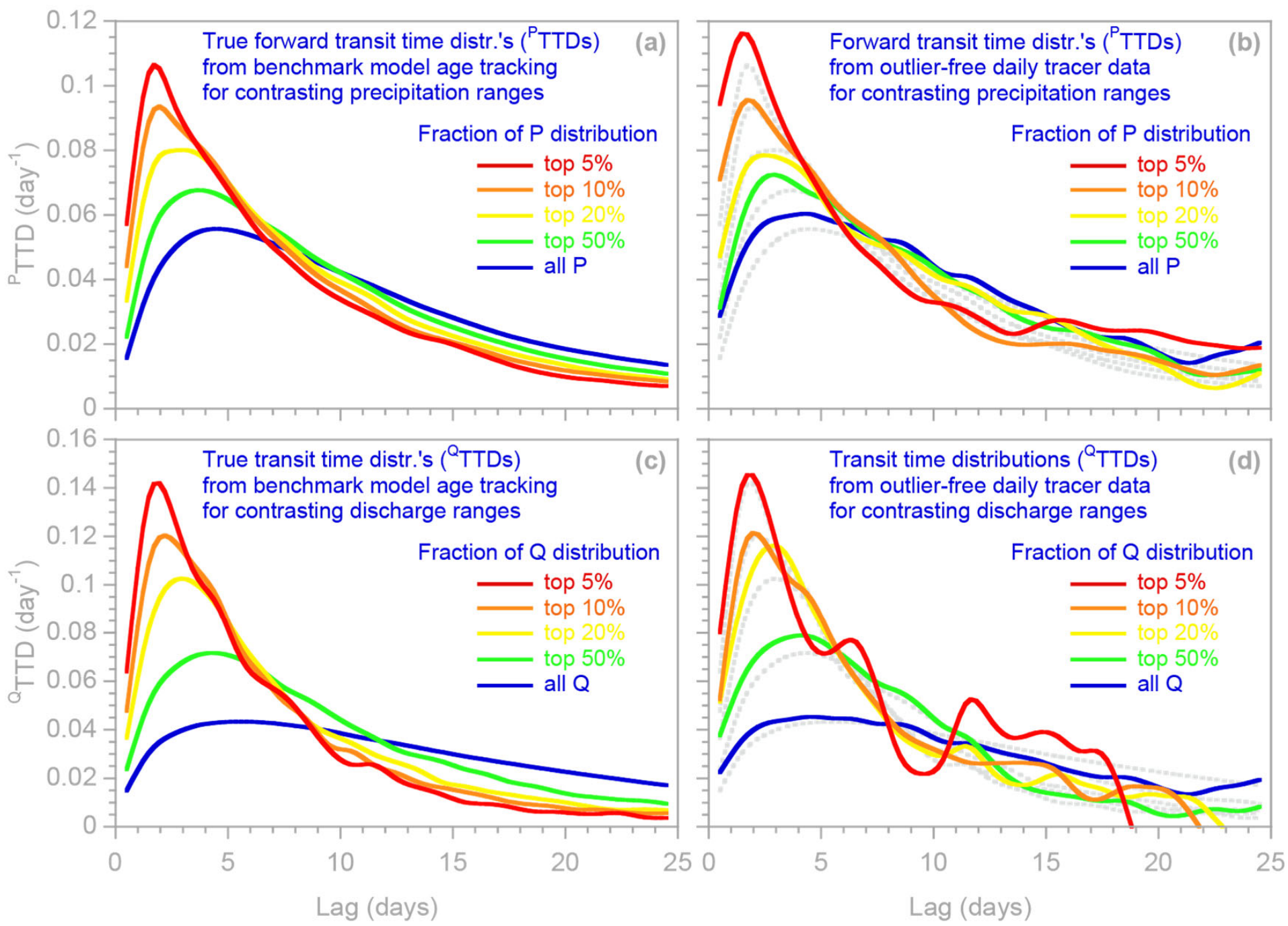

**Figure 11.** Non-stationary transit time distributions of precipitation (top panels) and discharge (bottom panels), visualized through age tracking in the benchmark model (left panels) and ensemble hydrograph separation (right panels) for selected ranges of precipitation and discharge in the daily time series. Dotted gray lines in right panels show model age tracking results (from left panels) for comparison. Curves show spline interpolations between individual points at each daily lag. The ensemble hydrograph separation TTDs differ somewhat from the model age tracking results, but they both exhibit similar progressions toward higher, earlier, and narrower TTD peaks at higher precipitation and discharge rates.

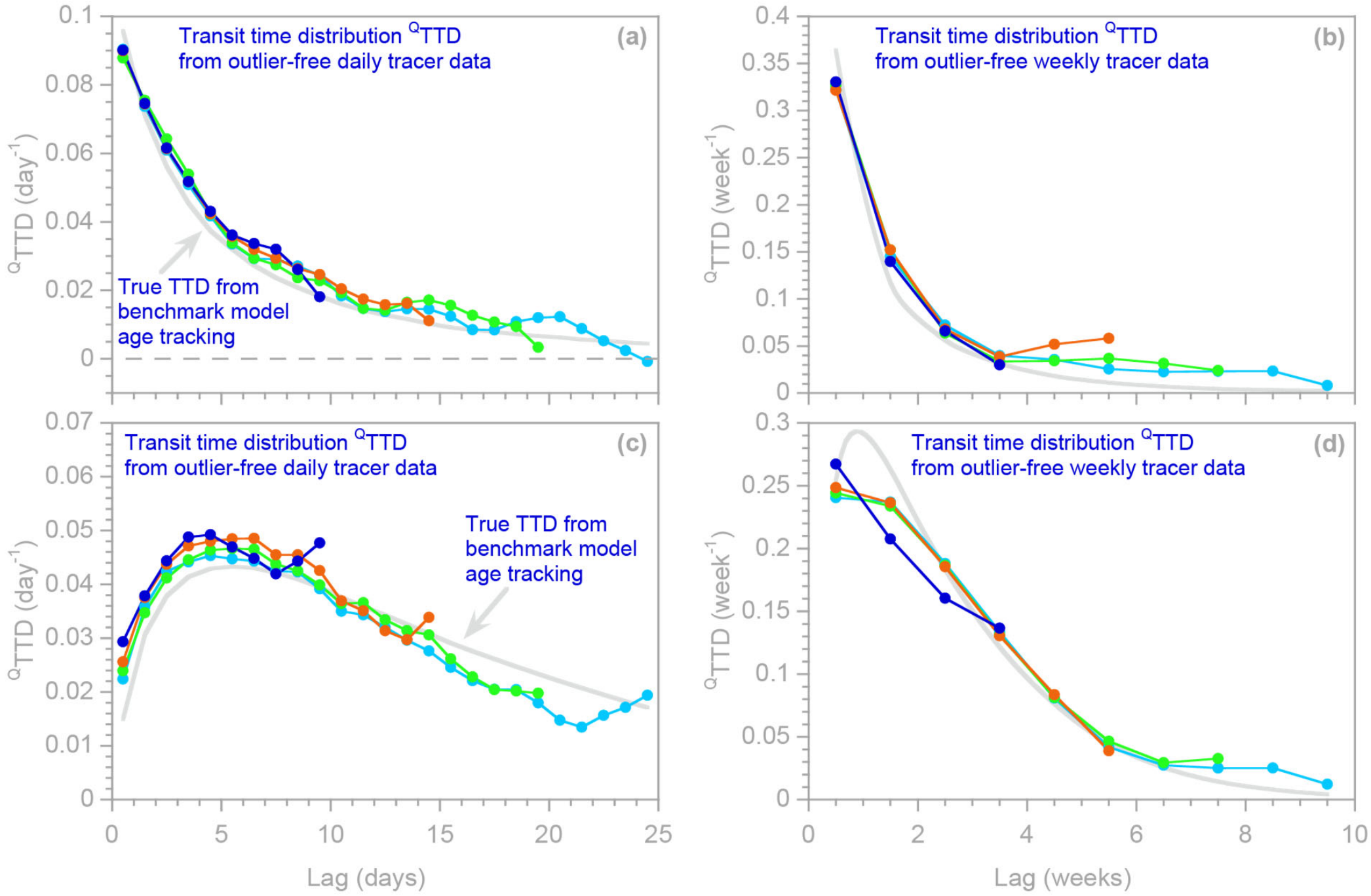

**Figure 12.** L-shaped (a, b) and humped (c, d) transit time distributions ($^Q$TTDs) calculated from benchmark daily (a, c) and weekly (b, d) tracer time series for different numbers of lag intervals (shown by the different colors in each panel). The light gray curves indicate the true TTDs, as calculated from water age tracking in the benchmark model. Standard errors are not shown to avoid obscuring the patterns in the overlapping TTD estimates. TTD estimates with different numbers of lags generally agree, except for their last few lags. The unusual case of the 4-lag TTD shown in dark blue in panel (d).