# Peer review of "Technical note: Calculation scripts for ensemble hydrograph separation"

_Hydrology and Earth System Sciences, 2020_

## Referee Comment (RC1) · Anonymous Referee #1 · 14 Sep 2020

The authors have developed a MATLAB script and an R script that estimates new water fractions and transit time distributions (TTDs) based on Kirchner's 2019 method. The method has been extended in this manuscript to provide robust estimations when outliers are present. I believe that this manuscript can serve as a good manual for potential users of that script. They have also provided some thoughtful analyses that would help users understand the potential limitations of the method.

The manuscript is well-written and mostly ready for publication. I only have some minor comments to help increase readability. Also, as a potential user, I have a few questions for the authors on how to use the method correctly.

[Figure]

**1. Required number of samples**

Could you suggest, at least, a rule of thumb number for the minimum number of samples required to perform an analysis using this method? I do not think that there would be a definite answer, and I guess it would depend on which analysis a user wants to do (among many others). Still, any suggestion would help potential users design their sampling strategy for their analysis of interest.

**2. New water fractions and TTDs estimations when the TTDs are humped**

The authors showed that the method overestimates uncertainty associated with the estimated averaged TTD when TTDs are humped. They argued that nonstationarity (time variability) of the TTDs might have caused the overestimation problem. If that is the case, is it possible to get better uncertainty estimations when one estimates TTDs for each subset (assuming that the subsets are well constructed)?

The authors also showed that the method overestimates the new water fraction at the daily time scale when the TTDs are humped. While they have shown that the issue can be resolved at the weekly time scale, I think that there is a way to get a good estimation at the daily time scale. Some of their explanations about the overestimation of the new water fraction and the results that are shown in Figure 6 imply that the method could estimate $F_{new}$ pretty well at the daily time scale if one estimates TTDs first (probably with $m$ about 7 days in this case, and for each subset to alleviate the uncertainty overestimation issue) and then use $\beta_0$ for $^Q F_{new}$?

**3. On the use of IRLS**

The role of IRLS is a bit unclear. Their robust estimation method consists of two steps (the MAD-based filtering and the use of IRLS), but those steps' relative importance is not discussed. As the authors described in lines 173-178, IRLS could be an additional source of getting less accurate estimates. Would it be possible that, in some cases, the method estimates better TTDs and new water fractions when only the filtering is

applied? Then, I think it would be great to provide an option to do the MAD-based filtering separately.

**4. Clarifications**

L9, L65: I am not sure if the method can "measure" TTDs and new water fractions.

L58: It is hard to understand why the strongly biased outliers are harder to detect and eliminate.

L61: "Large enough" – Wouldn't it makes the outliers easy to detect?

L317: The authors have used the term "nonstationarity" frequently throughout the manuscript. If I understand correctly, I think it should be "time variability," not non-stationarity.

L330: Perhaps better to provide the lag-1 serial correlation $r_{sc}$ for the non-humped TTD cases.

Figure 2: $C_P$ and $C_Q$ notations here do not match with the notation used in the text. In the text, the double subscript notation is used.

Figure 2b: Coloring the corrupted data point (using different colors for the corrupted $C_P$ and $C_Q$) would make the figure easier to understand.

---

## Referee Comment (RC2) · Anonymous Referee #2 · 15 Sep 2020

The Technical Note: "Calculation scripts for ensemble hydrograph separation" by Kirchner and Knapp, presents an ensemble hydrograph separation tool, useful to estimates new water fractions and transit time distributions (TTDs). The authors developed user-friendly scripts that perform EHS calculations in two broadly used platforms (MATLAB and R).

The authors used an impressive synthetic data set, that despite the limitations they clearly stated in the manuscript, mimics reasonably the real word behavior of isotope time series.

The authors made an important contribution to the scientific community by helping to

solve the common problem of lack of monitored/non-stationary end end-members while performing hydrograph separation. Moreover, they put great effort into describing the method, providing examples, and addressing uncertainties issues. I was delight by reading this technical note that is well-structured and clearly written.

Some of my main suggestions matched those of Reviewer RC1 (specifically related to IRLS and the overestimation of Fnew when the TTDs are humped) and were already clarified by the authors by including them as supplementary material.

I found this work in very good form and suggest the Editor accept this publication after a single additional clarification.

L 380-392 Could the authors please further explain the mismatch between the discharge age tracking using the benchmark model and the new ensemble hydrograph separation? As well as the potential implications for sampling size and frequency. This will be useful for users who will apply the method with real-world data.

---

## Author Comment (AC1) · 15 Sep 2020

**We thank the reviewer for these comments on our manuscript. Below we respond (in bold type) to the reviewer's comments (in normal type).**

The authors have developed a MATLAB script and an R script that estimates new water fractions and transit time distributions (TTDs) based on Kirchner's 2019 method. The method has been extended in this manuscript to provide robust estimations when outliers are present. I believe that this manuscript can serve as a good manual for potential users of that script. They have also provided some thoughtful analyses that would help users understand the potential limitations of the method. The manuscript is well-written and mostly ready for publication. I only have some minor comments to help increase readability. Also, as a potential user, I have a few questions for the authors on how to use the method correctly.

1. Required number of samples

Could you suggest, at least, a rule of thumb number for the minimum number of samples required to perform an analysis using this method? I do not think that there would be a definite answer, and I guess it would depend on which analysis a user wants to do (among many others). Still, any suggestion would help potential users design their sampling strategy for their analysis of interest.

**The required number of samples will indeed depend on many other factors beyond just what analysis the user wants to do (new water fraction vs. TTD). Some of those factors include:**

**a) how variable are the tracer concentrations in precipitation, and over what timescales?**

**b) how variable are the tracer concentrations in streamflow, and over what timescales? (Note that this will depend not only on the answer to (a), but also on the timescales of catchment storage – in other words, what the transit time distribution is.)**

**c) how stationary vs. non-stationary (time-invariant vs. time-variant) is the catchment's transit time distribution?**

**d) how large are the measurement uncertainties in the tracer concentrations? Are the measurement errors serially correlated, and by how much?**

**e) what errors or uncertainties in the results (Fnew and TTD values) are acceptable?**

**Many of these factors will be unknown in advance (for example, the sample size needed to estimate a TTD will vary, depending on what that TTD is, which will not be known – that's the whole point of estimating the TTD in the first place). Thus it is difficult at this stage to provide a rule of thumb, until we (and the community) gain more experience with real-world applications.**

**In the meantime, the most informative approach is to generate benchmark data sets under a range of assumptions, and then test how the sample size affects the accuracy of the inferred Fnew and TTD. Unfortunately we cannot recommend a way to short-cut that process.**

2. New water fractions and TTDs estimations when the TTDs are humped

The authors showed that the method overestimates uncertainty associated with the estimated averaged TTD when TTDs are humped. They argued that nonstationarity (time variability) of the

TTDs might have caused the overestimation problem. If that is the case, is it possible to get better uncertainty estimations when one estimates TTDs for each subset (assuming that the subsets are well constructed)?

**We already tried this and unfortunately it doesn't work. Or rather, it might theoretically work, but not under conditions that are likely to be encountered in the real world. Consider a simplified nonstationary system that has two different states, "wet", and "dry", with a different (stationary) TTD in each of these two states. If the "wet" state lasted long enough that the catchment stayed wet between the time the tracer entered the catchment in rainfall and exited in streamflow, and likewise if the "dry" state lasted long enough that the catchment stayed dry between tracers entering in precipitation and exiting in streamflow… and if one could cleanly split the data set between the "wet" and "dry" subsets, then yes, the strategy described by the reviewer could potentially work.**

**But this would require that the timescales over which the catchment switches between wet and dry conditions were much longer than the timescales over which the catchment stores tracers, which will rarely be the case. In the messy real world, by contrast, many different precipitation events, and many changes in catchment conditions, are overprinted on each other between the time that tracers enter in precipitation and leave in streamflow.**

The authors also showed that the method overestimates the new water fraction at the daily time scale when the TTDs are humped. While they have shown that the issue can be resolved at the weekly time scale, I think that there is a way to get a good estimation at the daily time scale. Some of their explanations about the overestimation of the new water fraction and the results that are shown in Figure 6 imply that the method could estimate Fnew pretty well at the daily time scale if one estimates TTDs first (probably with m about 7 days in this case, and for each subset to alleviate the uncertainty overestimation issue) and then use Beta_0 for QFnew?

**We already thought of this and already tried it, and the reason we didn't describe it is because it didn't work. (Indeed if we mentioned every intuitive-sounding idea that doesn't work, and explained why it doesn't work, the paper would be many times its present length.)**

3. On the use of IRLS

The role of IRLS is a bit unclear. Their robust estimation method consists of two steps (the MAD-based filtering and the use of IRLS), but those steps' relative importance is not discussed. As the authors described in lines 173-178, IRLS could be an additional source of getting less accurate estimates. Would it be possible that, in some cases, the method estimates better TTDs and new water fractions when only the filtering is applied? Then, I think it would be great to provide an option to do the MAD-based filtering separately.

**We really don't think this would be a good idea. MAD-based filtering and IRLS, like any other robust estimation procedures, will both reduce the accuracy of any results that rely on extreme values that are not actually outliers (but instead, for example, are simply the very long tails of an outlier-free distribution). If, on the other hand, the extreme values are indeed outliers, then these procedures will greatly improve the accuracy of the results (relative to those from non-robust analyses corrupted by outliers.**

**The only case in which it would make sense to used MAD-based filtering and not use IRLS would be if we knew that _all_ of the outliers were big enough to be detected and removed by MAD-based**

**filtering, and _none_ were small enough to get through the MAD filter. Such a situation seems highly improbable. Thus the decision to use robust estimation or not is, in our view, an either/or decision that the user should make.**

4. Clarifications

L9, L65: I am not sure if the method can "measure" TTDs and new water fractions.

**We can say "estimate" instead since the TTD and Fnew are not measured directly.**

**We will note, though, that many quantities that are actually estimated from proxies are typically called measurements instead. For example, an altimeter actually _measures_ air pressure, and uses it to calculate (or infer) altitude. A GPS unit actually _measures_ the relative arrival times of radio waves from GPS satellites, and calculates or infers the user's position. But most people have no problem saying that an altimeter measures altitude and a GPS measures one's location.**

L58: It is hard to understand why the strongly biased outliers are harder to detect and eliminate.

**Strongly biased outliers are harder to detect and eliminate because they shift the mean of the distribution, making it harder to distinguish between the outliers and the un-corrupted data. Data with strongly biased outliers may also be difficult to distinguish from the naturally skewed distributions that characterize many environmental variables.**

L61: "Large enough" – Wouldn't it makes the outliers easy to detect?

**That depends on the detection technique. The example shown in Fig. 2 involves some outliers that have so much leverage on the fitted line that it lies close to them – and thus they are harder to detect as outliers based on their residuals (which is why we can't rely on IRLS alone to do the job, since IRLS is based on identifying unusually large outliers).**

L317: The authors have used the term "nonstationarity" frequently throughout the manuscript. If I understand correctly, I think it should be "time variability," not nonstationarity.

**Nonstationarity refers, in conventional usage, to the time-variability of the statistical properties of a quantity (typically a distribution). Thus "nonstationary" and "time-varying" are typically used interchangeably. To make this equivalence explicit, we will add "(i.e., time-varying)" after one use of "nonstationary" and "(i.e., time-invariant)" after one use of "stationary".**

L330: Perhaps better to provide the lag-1 serial correlation rsc for the non-humped TTD cases.

**The goal of this analysis is to show how the non-stationarity of the humped TTDs leads to inflated standard errors. To show this, we compare results from stationary and non-stationary benchmark models that have similar average TTDs. The stationary and nonstationary benchmark models have the same parameter values, but one has constant precipitation (giving a stationary TTD) and the other has time-varying precipitation (giving a nonstationary TTD). If instead we compared benchmark models with different parameters, as suggested by the reviewer, we would not be able to demonstrate the role that non-stationarity plays in generating large standard errors.**

Figure 2: CP and CQ notations here do not match with the notation used in the text. In the text, the double subscript notation is used.

**That's a formatting issue in the plotting program, which doesn't allow double subscripts.  We'll fix it by hand.**

Figure 2b: Coloring the corrupted data point (using different colors for the corrupted CP and CQ) would make the figure easier to understand.

**Good point.  We will re-plot the figure with the same colors shown in Fig. 2a, and the outliers in black.**

---

## Author Comment (AC2) · 21 Sep 2020

**We thank reviewer #2 for these comments on our manuscript. Below we respond (in bold type) to the reviewer's comments (in normal type).**

The Technical Note: "Calculation scripts for ensemble hydrograph separation" by Kirchner and Knapp, presents an ensemble hydrograph separation tool, useful to estimates new water fractions and transit time distributions (TTDs). The authors developed user-friendly scripts that perform EHS calculations in two broadly used platforms (MATLAB and R).

The authors used an impressive synthetic data set, that despite the limitations they clearly stated in the manuscript, mimics reasonably the real word behavior of isotope time series.

The authors made an important contribution to the scientific community by helping to solve the common problem of lack of monitored/non-stationary end end-members while performing hydrograph separation. Moreover, they put great effort into describing the method, providing examples, and addressing uncertainties issues. I was delight by reading this technical note that is well-structured and clearly written.

Some of my main suggestions matched those of Reviewer RC1 (specifically related to IRLS and the overestimation of Fnew when the TTDs are humped) and were already clarified by the authors by including them as supplementary material.

I found this work in very good form and suggest the Editor accept this publication after a single additional clarification.

**Thanks very much for these comments.**

L 380-392 Could the authors please further explain the mismatch between the discharge age tracking using the benchmark model and the new ensemble hydrograph separation? As well as the potential implications for sampling size and frequency. This will be useful for users who will apply the method with real-world data.

**As we indicated in our response to reviewer #1, it is difficult to generalize here. What we show in Fig. 11, and comment on in lines 380-392, are results for one specific set of benchmark model parameters, and for the particular time series of precipitation fluxes and tracer concentrations shown in Fig. 1. It would of course be interesting to undertake a more systematic exploration of how the sample size and frequency, as well as the various benchmark model parameters and the characteristics of the precipitation time series, all affect the uncertainties and errors in ensemble hydrograph separation estimates.**

**But that would be an entirely different (and probably much longer) paper. The point of this paper is to make the codes for the method publically available and to briefly illustrate their possible uses and limitations. We should also keep this matter in perspective: both in the earlier paper (Kirchner, 2019) and in the present manuscript, we have already tested this method more rigorously than many others have been tested.**

**What we can do is to expand the discussion in Section 4.5 to also include some comments on sample size, based on our recent experience applying ensemble hydrograph separation to real-world data from Plynlimon (Knapp et al., 2019). In the revised manuscript, we will therefore begin Section 4.5 with the following:**

"Prospective users of ensemble hydrograph separation may naturally wonder what sample sizes and sampling frequencies are needed to estimate new water fractions and transit time distributions. The answers will depend on many different factors, including the time scales of interest to the user, the desired precision of the F_new and TTD estimates, the logistical constraints on sampling and analysis, the frequency and intermittency of precipitation events, the variability of the input tracers over different time scales, and the time scales of storage and transport in the catchment itself (that is, what the TTD is and how non-stationary it is, which of course can only be guessed before measurements are available). Ideally one should sample at a frequency that is high enough to capture the shortest time scales of interest, and sample much longer than the longest time scales of interest. One should also aim to capture many diverse transport events, spanning many different catchment conditions and precipitation characteristics.

Beyond these generalizations, it is difficult to offer concrete advice. We can, however, report our recent experience applying ensemble hydrograph separation to weekly and 7-hourly isotope time series at Plynlimon, Wales (Knapp et al., 2019). We were generally able to estimate TTDs out to lags of about three months based on four years of weekly sampling. The same four years of weekly samples yielded about 100 precipitation-discharge sample pairs (after samples corresponding to below-threshold precipitation were removed), which were sufficient to estimate weekly event new water fractions with an uncertainty of about 1% (e.g., $^{Qp}F_{new} \sim 8\pm1\%$). When these were split into four seasons, we could estimate event new water fractions with an uncertainty of about 2-3% using 20-30 weekly precipitation-discharge pairs, and when they were split into 4-6 different ranges of precipitation and discharge, we could reasonably well constrain the profiles of new water response to catchment wetness and precipitation intensity (Fig. 10 of Knapp et al., 2019). We were able to estimate 7-hourly TTDs out to lags of 7 days based on about 17 months of 7-hourly isotope samples, including almost 1500 discharge samples and 540 above-threshold precipitation samples, and splitting these data sets in half allowed us to distinguish the TTDs for summer and winter conditions (Figs. 11 and 12 of Knapp et al., 2019). However, these numbers should not be uncritically adopted as rules of thumb for other catchments, since precipitation at Plynlimon is frequent and weakly seasonal, and the catchment is characterized by rapid hydrological response but relatively long storage timescales (Kirchner et al., 2000). All of these characteristics could potentially affect the sample sizes needed for estimating new water fractions and transit time distributions. As more experience is gained at more catchments, general rules of thumb may emerge. Until then, however, benchmark tests like those described here can potentially provide a more reliable site-specific guide to sample size requirements."